# DEPTH ANYTHING WITH ANY PRIOR

**Zehan Wang**[1][*], **Siyu Chen**[1][*], **Lihe Yang**[2], **Jialei Wang**[1],
**Ziang Zhang**[1] , **Hengshuang Zhao**[2], **Zhou Zhao**[1][†]

[1]Zhejiang University; [2]The University of Hong Kong
`https://prior-depth-anything.github.io/`

## ABSTRACT

This work presents ***Prior Depth Anything***, a framework that combines incomplete but precise metric information in depth measurement with relative but complete geometric structures in depth prediction, generating accurate, dense, and detailed metric depth maps for any scene. To this end, we design a coarse-to-fine pipeline to progressively integrate the two complementary depth sources. First, we introduce pixel-level metric alignment and distance-aware weighting to pre-fill diverse metric priors by explicitly using depth prediction. It effectively narrows the domain gap between prior patterns, enhancing generalization across varying scenarios. Second, we develop a conditioned monocular depth estimation (MDE) model to refine the inherent noise of depth priors. By conditioning on the normalized pre-filled prior and prediction, the model further implicitly merges the two complementary depth sources. Our model showcases impressive zero-shot generalization across depth completion, super-resolution, and inpainting over 7 real-world datasets, matching or even surpassing previous task-specific methods. More importantly, it performs well on challenging, unseen mixed priors and enables test-time improvements by switching prediction models, providing a flexible accuracy-efficiency trade-off while evolving with advancements in MDE models.

## 1 INTRODUCTION

Fine-detailed and dense metric depth information is a fundamental pursuit in computer vision and robotics applications (Zhang et al., 2023a; Deng et al., 2022; Chung et al., 2024; Roessle et al., 2022; Esser et al., 2023; Wang et al., 2024; Zhen et al., 2024; Wang et al., 2019; Wofk et al., 2019; Wang et al., 2025b; Zhang et al., 2025; Wang et al., 2025c; 2023; Huang et al., 2024). Although monocular depth estimation (MDE) models (Ranftl et al., 2020; Yang et al., 2024a;b; Ke et al., 2024; Bochkovskii et al., 2025; He et al., 2025; Hu et al., 2024; Yin et al., 2023) have made

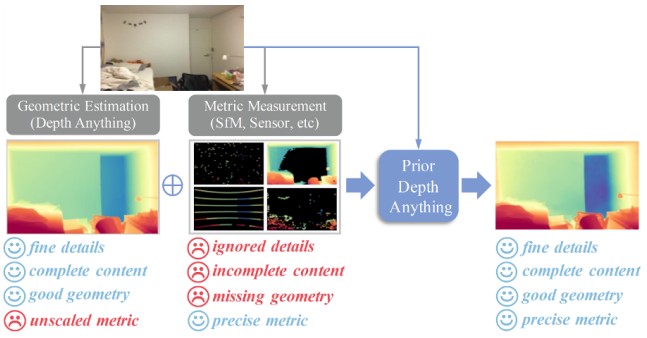

Figure 1: **Core Motivation**. We progressively integrate complementary information from metric measurements (accurate metrics) and relative predictions (completeness and fine details) to produce dense and fine-grained metric depth maps.

significant progress, enabling complete and detailed depth predictions, predicted depths are relative and lack precise metric information. On the other hand, depth measurement technologies, such as Structure from Motion (SfM) (Schonberger & Frahm, 2016) or depth sensors (Lange & Seitz, 2001), provide precise but often incomplete and coarse metric information.

---

[*]Equal Contribution.
[†]Corresponding Author.

| Methods | Target Task | Sparse Point | | | Low-resolution | Missing Area | | | Mixed |
|---|---|---|---|---|---|---|---|---|---|
| | | *SfM* | *LiDAR* | *Extreme* | | *Range* | *Shape* | *Object* | |
| Marigold-DC (Viola et al., 2024) | Depth Completion | ✓ | ✓ | ✓ | ✗ | ✗ | ✗ | ✗ | ✗ |
| Omni-DC (Zuo et al., 2024) | Depth Completion | ✓ | ✓ | ✓ | ✓ | ✗ | ✗ | ✗ | ✗ |
| PromptDA (Lin et al., 2025) | Depth Super-resolution | ✗ | ✗ | ✗ | ✓ | ✗ | ✗ | ✗ | ✗ |
| DepthLab (Liu et al., 2024a) | Depth Inpainting | ✗ | ✗ | ✗ | ✗ | ✗ | ✓ | ✓ | ✗ |
| Prior Depth Anything | All-Rounder | ✓ | ✓ | ✓ | ✓ | ✓ | ✓ | ✓ | ✓ |

Table 1: Applicable scenarios of current prior-based monocular depth estimation models. *SfM*: sparse matching points from SfM, *LiDAR*: sparse LiDAR line patterns, *Extreme*: extremely sparse points (100 points), *Range*: missing depth within a specific range, *Shape*: missing regular-shaped areas, *Object*: missing depth of an object.

In this paper, we explore *prior-based monocular depth estimation*, which takes RGB images and measured depth priors as inputs to output detailed and precise metric depth maps. We unify different depth estimation tasks by abstracting various types of depth measurements as depth prior. To clarify the scope, we first outline common types of depth priors and their primary applications:

**1) Sparse points (depth completion):** Depth from LiDAR or SfM (Schonberger & Frahm, 2016) is typically sparse. Completing sparse depth priors is crucial for applications such as 3D reconstruction (Roessle et al., 2022; Chung et al., 2024) and autonomous driving (Tao et al., 2022; Häne et al., 2017; Carranza-García et al., 2022). **2) Low-resolution (depth super-resolution):** Low-power Time of Flight (ToF) cameras (Lange & Seitz, 2001), commonly used in mobile phones, capture low-resolution depth maps. Depth super-resolution is essential for spatial perception, VR (Rasla & Beyeler, 2022), and AR (Slater & Wilbur, 1997) in portable devices (He et al., 2021a). **3) Missing areas (depth inpainting):** Stereo matching failures (Lowe, 2004; Rublee et al., 2011) or 3D Gaussian splatting edits (Liu et al., 2024b; Yu et al., 2024) may leave large missing areas in depth maps. Filling these gaps is vital for 3D scene generation and editing (Yu et al., 2024). **4) Mixed prior:** In real-world scenarios, different depth priors often coexist. For instance, structured light cameras (Herrera et al., 2012) often generate low-resolution and incomplete depth maps, due to their limited working range. Handling these mixed priors is vital for practical applications.

In Tab. 1, we detail the patterns of each prior. Existing methods mainly focus on specific limited priors, limiting their use in diverse, real-world scenarios. In this work, we propose *Prior Depth Anything*, motivated by the complementary advantage between predicted and measured depth maps, as illustrated in Fig. 1. Technically, we design a coarse-to-fine pipeline to explicitly and progressively combine the depth prediction with measured depth prior, achieving impressive robustness to any image with any prior.

We first introduce coarse metric alignment to pre-fill incomplete depth priors using predicted relative depth maps, which effectively narrows the domain gap between various prior types. Next, we apply the fine structure refinement to rectify misaligned geometric structures in the pre-filled depth priors caused by inherent noises in the depth measurements. Specifically, the pre-filled depth prior (with accurate metric data) and the relative depth prediction (with fine details and structure) are provided as additional inputs to a conditioned MDE model. Guided by the RGB image input, the model can combine the strengths of two complementary depth sources for the final output.

We evaluate our model on 7 datasets with varying depth priors. It achieves zero-shot depth completion, super-resolution, and inpainting within a single model, matching or outperforming previous models that are specialized for only one of these tasks. More importantly, our model achieves significantly better results when different depth priors are mixed, highlighting its effectiveness in more practical and varying scenarios.

Our contribution can be summarized as follows:

- We propose *Prior Depth Anything*, a unified framework to estimate fine-detailed and complete metric depth with any depth priors. Our model can seamlessly handle zero-shot depth completion, super-resolution, inpainting, and adapt to more varied real-world scenarios.

- We introduce coarse metric alignment to pre-fill depth priors, narrowing the domain gap between different types of depth prior and enhancing the model's generalization.

- We design fine structure refinement to alleviate the inherent noise in depth measurements. This involves a conditioned MDE model to granularly merge the pre-filled depth prior and prediction based on image content.

- Our method exhibits superior zero-shot results across various datasets and tasks, surpassing even state-of-the-art methods specifically designed for individual tasks.

## 2 RELATED WORK

### 2.1 MONOCULAR DEPTH ESTIMATION

Monocular depth estimation (MDE) is a fundamental computer vision task that predicts the depth of each pixel from a single color image (Eigen et al., 2014; Fu et al., 2018; Bhat et al., 2021). Recently, with the success of "foundation models" (Bommasani et al., 2021), some studies (Ranftl et al., 2020; Yang et al., 2024a;b; Xu et al., 2025; He et al., 2025; Ke et al., 2024; Bochkovskii et al., 2025; Yin et al., 2023; Xinlai et al., 2026) have attempted to build depth foundation models by scaling up data and using stronger backbones, enabling them to predict detailed geometric structures for any image.

MiDaS (Ranftl et al., 2020) made the pioneering study by training an MDE model on joint datasets to improve generalization. Following this line, Depth Anything v1 (Yang et al., 2024a) scaled training with massive unlabeled image data, while Depth Anything v2 (Yang et al., 2024b) further enhanced its ability to handle fine details, reflections, and transparent objects by incorporating highly precise synthetic data (Wang et al., 2021; 2020; Yao et al., 2020; Cabon et al., 2020; Roberts et al., 2021).

Although these methods have demonstrated high accuracy and robustness, they primarily produce unscaled relative depth maps due to the significant scale differences between indoor and outdoor scenes. While Metric3D (Yin et al., 2023; Hu et al., 2024) and Depth Pro (Bochkovskii et al., 2025) achieve zero-shot metric depth estimation through canonical camera transformations, the precision remains limited compared to measurement technologies.

Our method builds on the strength of existing depth foundation models, which excel at precisely capturing relative geometric structures and fine details in any image. By progressively integrating accurate but incomplete metric information in the depth measurements, our model can generate dense and detailed metric depth maps for any scene.

### 2.2 PRIOR-BASED MONOCULAR DEPTH ESTIMATION

In practical applications, depth measurement methods like multi-view matching (Cordts et al., 2016) or sensors (Silberman et al., 2012; Geiger et al., 2013) can provide accurate metric information, but due to their inherent nature or cost limitations, these measurements often capture incomplete information. Some recent studies have tried to use this measurement data as prior knowledge in the depth estimation process to achieve dense and accurate metric depth. These methods, however, primarily focus on specific patterns of depth measurement, which can be categorized into three types based on their input patterns:

**Depth Completion** As noted in (Roessle et al., 2022), SfM reconstructions from 19 images often result in depth maps with only 0.04% valid pixels. Completing the sparse depth maps with observed RGB images is a fundamental computer vision task (Tang et al., 2024; Cheng et al., 2020; 2019; Zuo & Deng, 2024; Zhang et al., 2023b; Park et al., 2024). Recent approaches like Omni-DC (Zuo et al., 2024) and Marigold-DC (Viola et al., 2024) have achieved certain levels of zero-shot generalization across diverse scenes and varying sparsity levels. However, due to the lack of explicit scene geometry guidance, they face challenges in extremely sparse scenarios.

**Depth Super-resolution** Obtaining high-resolution metric depth maps with depth cameras usually demands significant power. A more efficient alternative is to use low-power sensors to capture low-resolution maps and then enhance them using super-resolution. Early efforts (Zhong et al., 2023; Xian et al., 2020; Zhao et al., 2022; He et al., 2021b), however, show limited generalization to unseen scenes. Recent PromptDA (Lin et al., 2025) achieves effective zero-shot super-resolution by using the low-resolution map as a prompt for depth foundation models (Yang et al., 2024b).

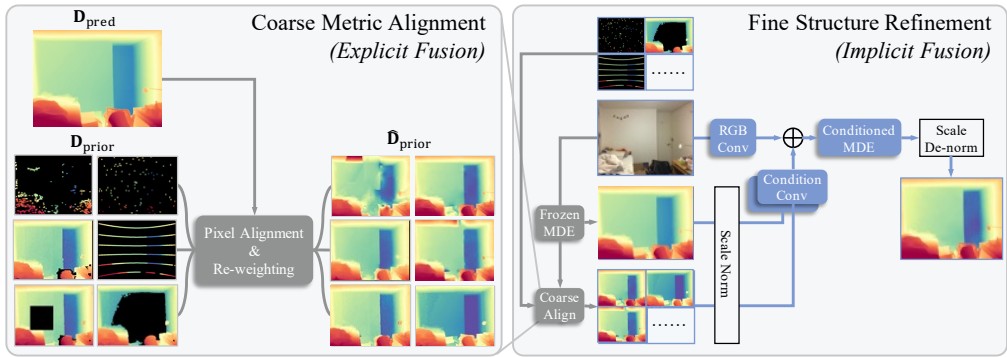

Figure 2: ***Prior Depth Anything.*** Considering RGB images, any form of depth prior $\mathbf{D}_{\text{prior}}$, and relative prediction $\mathbf{D}_{\text{pred}}$ from a frozen MDE model, coarse metric alignment first explicitly combines the metric data in $\mathbf{D}_{\text{prior}}$ and geometry structure in $\mathbf{D}_{\text{pred}}$. Fine structure refinement implicitly merges the complementary information to produce the final metric depth map.

**Depth Inpainting** As discussed in (Yang et al., 2024b; Ke et al., 2024), due to inherent limitations in stereo matching and depth sensors, even "ground truth" depth data in real-world datasets often have significant missing regions. Additionally, in applications like 3D Gaussian editing and generation (Liu et al., 2024b; Yu et al., 2024; Chung et al., 2023), there is a need to fill holes in depth maps. DepthLab (Liu et al., 2024a) first fills holes using interpolation and then refines the results with a depth-guided diffusion model. However, interpolation errors reduce its effectiveness for large missing areas or incomplete depth ranges.

These previous methods have two main limitations: 1) Poor performance when prior is limited. 2) Difficulty generalizing to unseen prior patterns. Our approach, Prior Depth Anything, tackles these challenges by explicitly using geometric information from depth prediction in a coarse-to-fine process, achieving impressive generalization and accuracy across various patterns of prior input.

## 3 PRIOR DEPTH ANYTHING

Advanced monocular depth estimation models excel in predicting detailed, complete relative depth maps with precise geometric structures for any image. In contrast, depth measurement technologies can provide metric depth maps but suffer from inherent noise and varying patterns of incompleteness. Inspired by the complementary strengths of estimated and measured depth, we introduce ***Prior Depth Anything*** to progressively and effectively merge the two depth sources. To handle diverse real-world scenarios, we take measurement depth in any form as the metric prior, producing fine-grained and complete metric depth maps for any image with any prior.

### 3.1 PRELIMINARY

Given an RGB image $\mathbf{I} \in \mathbb{R}^{3 \times H \times W}$ and its corresponding metric depth prior $\mathbf{D}_{\text{prior}} \in \mathbb{R}^{H \times W}$, prior-based monocular depth estimation takes the $\mathbf{I}$ and $\mathbf{D}_{\text{prior}}$ as input, aiming to output the depth map $\mathbf{D}_{\text{output}} \in \mathbb{R}^{H \times W}$ that is detailed, complete, and metrically precise. We uniformly represent the coordinates of valid positions in $\mathbf{D}_{\text{prior}}$ as $\mathbf{P} = \{x_i, y_i\}_{i=0}^{N}$, which $N$ pixels are valid.

### 3.2 COARSE METRIC ALIGNMENT

As shown in Fig. 2, different types of depth priors exhibit distinct missing patterns (e.g. sparse points, low-resolution grids, or irregular holes). These differences in sparsity and incompleteness restrict models' ability to generalize across various priors. To tackle this, we propose pre-filling missing regions to transform all priors into a shared intermediate domain.

**Pixel-level Metric Alignment** We first use a frozen MDE model to obtain a relative depth prediction $\mathbf{D}_{\text{pred}} \in \mathbb{R}^{H \times W}$. Then, by explicitly utilizing the accurate geometric structure in the predicted depth, we fill the invalid regions in $\mathbf{D}_{\text{prior}}$ pixel by pixel. Considering the pre-filled coarse depth map $\hat{\mathbf{D}}_{\text{prior}}$, which inherits all the valid pixels in $\mathbf{D}_{\text{prior}}$:

$$\hat{\mathbf{D}}_{\text{prior}}(x, y) = \mathbf{D}_{\text{pred}}(x, y), \text{ where } (x, y) \in \mathbf{P} \tag{1}$$

For each missing pixel $(\hat{x}, \hat{y})$, we first identify its $k$-nearest neighborhood (kNN) points $\{x_k, y_k\}_{k=1}^{K}$ from the valid pixel set $\mathbf{P}$. Then, we compute the optimal scale $s$ and shift $t$ that minimizes the least-squares error between the depth values of $\mathbf{D}_{\text{pred}}$ and $\mathbf{D}_{\text{prior}}$ at the $K$ supporting points:

$$s, t = \arg\min_{s,t} \sum_{k=1}^{K} \|s \cdot \mathbf{D}_{\text{pred}}(x_k, y_k) + t - \mathbf{D}_{\text{prior}}(x_k, y_k)\|_2 \quad (2)$$

With the optimal scale $s$ and shift $t$, we fill the missing pixels in $\hat{\mathbf{D}}_{\text{prior}}$ by linearly aligning the predicted depth value at $(\hat{x}, \hat{y})$ to metric prior:

$$\hat{\mathbf{D}}_{\text{prior}}(\hat{x}, \hat{y}) = s \cdot \mathbf{D}_{\text{pred}}(\hat{x}, \hat{y}) + t \quad (3)$$

**Distance-aware Re-weighting** Although simple pixel-level metric alignment achieves reasonable accuracy and generalization, two limitations remain: 1) *Discontinuity Risk*: Adjacent pixels in missing regions may select different k-nearest neighbors, resulting in abrupt changes. 2) *Uniform Weighting*: Nearby supporting points offer more reliable metric cues than distant ones, but equal weighting in the least squares overlooks this geometrical correlation, leading to suboptimal alignment.

To handle this issue, we further introduce distance-aware weighting for more smooth and accurate alignment. Within the alignment objective of Eq. 2, we re-weight each supporting point based on its distance to the query pixel, modifying Eq. 2 to:

$$s, t = \arg\min_{s,t} \sum_{k=1}^{K} \frac{\|s \cdot \mathbf{D}_{\text{pred}}(x_k, y_k) + t - \mathbf{D}_{\text{prior}}(x_k, y_k)\|_2}{\|(\hat{x}, \hat{y}) - (x_k, y_k)\|_2} \quad (4)$$

This simple modification ensures smoother transitions between regions and improves robustness by emphasizing geometrically closer measurements.

In summary, by explicitly integrating the accurate metric information from $\mathbf{D}_{\text{prior}}$ and the fine geometric structures from $\mathbf{D}_{\text{pred}}$, we cultivate the pre-filled dense prior $\hat{\mathbf{D}}_{\text{prior}}$, which offers two main advantages: *1) Similar Pattern:* Filling the missing area narrows the differences between various prior types, improving the generalization across different scenarios. *2) Fine Geometry*: The filled regions, derived from linear transformations of depth prediction, natively preserve the fine geometric structures, significantly boosting performance when prior information is limited.

### 3.3 Fine Structure Refinement

Although the prefilled coarse dense depth is generally accurate in metric, the parameter-free approach is sensitive to noise in depth priors. A single noisy pixel on blurred edges can disrupt all filled regions relying on it as a supporting point. To tackle these errors, we further implicitly leverage the MDE model's ability in capturing precise geometric structures in RGB images, learning to rectify the noise in priors and produce refined depth.

**Metric Condition** Specifically, we incorporate the pre-filled prior $\hat{\mathbf{D}}_{\text{prior}}$ as extra conditions to the pre-trained MDE model. With the guidance of RGB images, the conditioned MDE model is trained to correct the potential noise and error in $\hat{\mathbf{D}}_{\text{prior}}$. To this end, we introduce a condition convolutional layer parallel to the RGB input layer, as shown in Fig. 2. By initializing the condition layer to zero, our model can natively inherit the ability of pre-trained MDE model.

**Geometry Condition** In addition to leveraging the MDE model's inherent ability in capturing geometric structures from RGB input, we also incorporate existing depth predictions as an external geometry condition to help refine the coarse pre-filled prior. The depth prediction $\mathbf{D}_{\text{pred}}$ from frozen MDE model, is also passed into the condition MDE model through a zero-initialized conv layer.

**Scale Normalization** Then, we normalize the metric condition $\hat{\mathbf{D}}_{\text{prior}}$ and geometry condition $\mathbf{D}_{\text{pred}}$ to [0,1] for two key benefits: *1) Better Scene Generalization*: different scenes (e.g. indoor vs. outdoor) have significant depth scale differences. Normalization removes this scale variance, improving performance across diverse scenes. *2) Better MDE Model Generalization*: predictions from different frozen MDE models also have varying scales. Normalizing $\mathbf{D}_{\text{pred}}$ enables test-time model

| Model | Encoder | NYUv2 | | | ScanNet | | | ETH-3D | | | DIODE | | | KITTI | | | Ave. Rank |
|---|---|---|---|---|---|---|---|---|---|---|---|---|---|---|---|---|---|
| | | S+M | L+M | S+L | S+M | L+M | S+L | S+M | L+M | S+L | S+M | L+M | S+L | S+M | L+M | S+L | |
| DAv2 | ViT-L | 4.59 | 4.95 | 5.08 | 4.34 | 4.55 | 4.62 | 6.82 | 10.99 | 7.49 | 11.57 | 10.23 | 12.46 | 10.20 | 11.12 | 10.97 | 7.5 |
| Depth Pro | ViT-L | 4.46 | 4.87 | 5.41 | 4.22 | 4.44 | 4.51 | 6.39 | 7.08 | 9.29 | 12.92 | 8.42 | 8.91 | 6.29 | 9.24 | 9.18 | 6.7 |
| Omni-DC | - | 2.86 | 3.26 | 3.81 | 2.88 | 2.76 | 3.64 | 2.09 | 4.17 | 4.59 | 4.23 | 4.80 | 5.40 | 4.36 | 8.63 | 9.02 | 4.2 |
| Marigold-DC | SDv2 | 2.26 | 3.38 | 3.82 | 2.19 | 2.87 | 3.37 | 2.15 | 4.77 | 5.13 | 4.98 | 6.73 | 7.25 | 5.82 | 9.67 | 10.05 | 5.1 |
| DepthLab | SDv2 | 6.33 | 3.96 | 5.80 | 5.16 | 3.38 | 5.24 | 7.87 | 4.70 | 7.45 | 8.83 | 6.40 | 8.62 | 39.29 | 23.12 | 30.96 | 7.1 |
| PromptDA | ViT-L | 17.00 | 3.76 | 17.13 | 15.27 | 4.21 | 15.44 | 18.34 | 9.01 | 18.73 | 18.24 | 9.97 | 18.55 | 21.61 | 54.35 | 22.14 | 8.4 |
| PriorDA (ours) | DAv2-B+ViT-S | 2.09 | 2.88 | 3.17 | 2.14 | 2.56 | 2.94 | 1.65 | 3.96 | 4.16 | 3.76 | 4.43 | 4.89 | 3.97 | 8.38 | 8.53 | 2.9 |
| | DAv2-B+ViT-B | 2.04 | 2.82 | 3.09 | 2.11 | 2.50 | 2.86 | 1.56 | 3.84 | 4.08 | 3.62 | 4.30 | 4.73 | 3.86 | 8.40 | 8.57 | 2.0 |
| | Depth Pro+ViT-B | 2.01 | 2.82 | 3.08 | 2.06 | 2.48 | 2.82 | 1.61 | 3.83 | 4.05 | 3.40 | 4.23 | 4.60 | 3.37 | 8.12 | 8.25 | 1.1 |

Table 2: **Zero-shot depth estimation with mixed prior.** All results are reported in AbsRel ↓. "S": "Extreme" setting in sparse points, "L": "×16" in low-Resolution, "M": "Shape" (square masks of 160) in missing area. "Depth Pro+ViT-B" indicates the frozen MDE and conditioned MDE. DAv2-B: Depth Anything v2 ViT-B (Yang et al., 2024b), SDv2: Stable Diffusion v2. (Rombach et al., 2022). We highlight **Best**, second best results.

| Model | Encoder | NYUv2 | | | ScanNet | | | ETH-3D | | | DIODE | | | KITTI | | | Ave. Rank |
|---|---|---|---|---|---|---|---|---|---|---|---|---|---|---|---|---|---|
| | | SfM | LiDAR | Extreme | SfM | LiDAR | Extreme | SfM | LiDAR | Extreme | SfM | LiDAR | Extreme | SfM | LiDAR | Extreme | |
| DAv2 | ViT-L | 5.31 | 4.85 | 4.77 | 5.88 | 4.60 | 4.68 | 5.84 | 7.41 | 6.61 | 12.45 | 12.55 | 14.37 | 9.83 | 8.86 | 9.25 | 7.3 |
| Depth Pro | ViT-L | 4.80 | 4.47 | 4.41 | 5.27 | 4.12 | 4.23 | 5.68 | 5.31 | 6.51 | 10.53 | 9.08 | 8.98 | 6.45 | 6.05 | 6.19 | 6.1 |
| Omni-DC | - | 2.87 | 2.12 | 2.63 | 6.09 | 2.02 | 2.71 | 2.57 | 1.88 | 1.98 | 4.99 | 3.96 | 4.13 | 3.34 | 5.27 | 4.17 | 3.7 |
| Marigold-DC | SDv2 | 2.65 | 1.90 | 2.13 | 4.32 | 1.76 | 2.12 | 4.65 | 2.27 | 2.03 | 8.41 | 5.12 | 4.77 | 6.19 | 6.88 | 5.62 | 4.3 |
| DepthLab | SDv2 | 5.92 | 4.30 | 6.30 | 9.87 | 3.56 | 5.09 | 13.82 | 6.40 | 8.01 | 16.67 | 7.45 | 8.66 | 25.91 | 37.17 | 40.29 | 7.6 |
| PromptDA | ViT-L | 18.68 | 17.59 | 16.96 | 18.13 | 15.99 | 15.18 | 25.02 | 18.86 | 18.18 | 25.46 | 18.58 | 17.93 | 22.26 | 21.96 | 21.39 | 8.8 |
| PriorDA (ours) | DAv2-B+ViT-S | 2.42 | 2.01 | 2.01 | 3.90 | 2.19 | 2.09 | 3.26 | 1.90 | 1.61 | 5.25 | 3.88 | 3.65 | 3.73 | 4.81 | 3.76 | 3.3 |
| | DAv2-B+ViT-B | 2.38 | 1.95 | 1.97 | 3.85 | 2.15 | 2.04 | 3.10 | 1.82 | 1.50 | 5.16 | 3.77 | 3.64 | 3.74 | 4.73 | 3.76 | 2.2 |
| | Depth Pro+ViT-B | 2.36 | 1.96 | 1.96 | 3.84 | 2.14 | 2.01 | 3.13 | 1.84 | 1.54 | 5.07 | 3.66 | 3.37 | 3.35 | 4.12 | 3.28 | 1.7 |

Table 3: **Zero-shot depth completion** (e.g. sparse points prior). All results are reported in AbsRel ↓. "SfM": points sampled with SIFT (Lowe, 2004) and ORB (Rublee et al., 2011) following Omni-DC (Zuo et al., 2024), "LiDAR": 8 LiDAR lines, "Extreme": 100 random points.

switching, offering flexible accuracy-efficiency trade-offs for diverse demands, and enabling seamless improvements as MDE models advance.

**Synthetic Training Data** As discussed in (Ke et al., 2024; Yang et al., 2024b), real depth datasets often face issues such as blurred edges and missing values. Therefore, we leverage synthetic datasets, Hypersim (Roberts et al., 2021) and vKITTI (Cabon et al., 2020), with precise GT to drive our conditioned MDE model to rectify the noise in measurements. From the precise ground truth, we randomly sample sparse points, create square missing areas, or apply downsampling to construct different synthetic priors. To mimic real-world measurement noise, we add outliers and boundary noise to perturb the sampled prior, following (Zuo et al., 2024).

**Learning Objective** As mentioned earlier, both the metric and geometry conditions are normalized. Thus, we apply the de-normalization transformation to convert the output into the ground truth scale. Following ZoeDepth (Bhat et al., 2023), we use the scale-invariant log loss for supervision.

# 4 EXPERIMENT

## 4.1 EXPERIMENTAL SETTING

**Implementation Details** During training, we utilize the Depth Anything V2 ViT-B model as the frozen MDE model to produce relative depth predictions. During inference, the frozen MDE model can be swapped with any other pre-trained model. The $k$-value of the $k$NN process in Sec. 3.2 is set to 5. We initialize the conditioned MDE Model with two versions of Depth Anything V2: ViT-S and ViT-B. We train the conditioned MDE model for 200K steps with a batch size of 64, using 8 GPUs. The AdamW optimizer with a cosine scheduler is employed, where the base learning rate is set to 5e-6 for the MDE encoder and 5e-5 for the MDE decoder.

**Benchmarks** Our method aims to provide accurate and complete metric depth maps in a zero-shot manner for any image with any prior. **To cover "any image"**, we evaluate models on 7 unseen real-world datasets, including NYUv2 (Silberman et al., 2012) and ScanNet (Dai et al., 2017) for indoor, ETH3D (Schops et al., 2017) and DIODE (Vasiljevic et al., 2019) for indoor/outdoor, KITTI (Geiger et al., 2012) for outdoor, ARKitScenes (Baruch et al., 2021) and RGB-D-D (He et al., 2021b) for cap-

| Model | Encoder | ARKitScenes | | RGB-D-D | | NYUv2 | | ScanNet | | ETH-3D | | DIODE | | KITTI | | Ave. Rank |
|---|---|---|---|---|---|---|---|---|---|---|---|---|---|---|---|---|
| | | AbsRel↓ | RMSE↓ | AbsRel↓ | RMSE↓ | 8× | 16× | 8× | 16× | 8× | 16× | 8× | 16× | 8× | 16× | |
| DAv2 | ViT-L | 3.67 | 0.0764 | 4.67 | 0.1116 | 4.77 | 5.13 | 4.64 | 4.85 | 6.27 | 7.38 | 12.49 | 11.20 | 9.54 | 11.22 | 8.9 |
| Depth Pro | ViT-L | 3.25 | 0.0654 | 4.28 | 0.1030 | 4.48 | 4.83 | 4.17 | 4.40 | 5.88 | 6.79 | 8.20 | 8.33 | 6.76 | 9.16 | 7.8 |
| Omni-DC | - | 2.14 | 0.0435 | 2.09 | 0.0685 | **1.57** | 3.11 | **1.29** | 2.65 | 1.86 | 4.09 | **2.81** | 4.71 | 4.05 | 8.35 | 3.9 |
| Marigold-DC | SDv2 | 2.17 | 0.0448 | 2.15 | 0.0672 | 1.83 | 3.32 | 1.63 | 2.83 | 2.33 | 4.75 | 4.28 | 6.60 | 5.17 | 9.47 | 6.2 |
| DepthLab | SDv2 | 2.10 | 0.0411 | 2.13 | 0.0624 | 2.60 | 3.73 | 1.89 | 3.19 | 2.60 | 4.50 | 4.42 | 6.16 | 17.17 | 22.90 | 6.4 |
| PromptDA | ViT-L | 1.34 | 0.0347 | 2.79 | 0.0708 | 1.61 | **1.75** | 1.87 | **1.93** | **1.80** | **2.56** | 3.18 | 3.73 | **3.92** | **4.95** | 2.6 |
| PriorDA (ours) | DAv2-B+ViT-S | 2.09 | 0.0414 | 2.07 | 0.0597 | 1.73 | 2.79 | 1.60 | 2.50 | 2.06 | 3.91 | 3.09 | 4.36 | 4.54 | 8.20 | 3.9 |
| | DAv2-B+ViT-B | **1.94** | **0.0404** | **2.02** | **0.0581** | 1.72 | 2.73 | 1.61 | 2.45 | 2.00 | 3.79 | 3.10 | 4.23 | 4.65 | 8.24 | 2.9 |
| | Depth Pro+ViT-B | 1.95 | 0.0408 | **2.02** | **0.0581** | 1.72 | 2.74 | 1.58 | 2.43 | 1.99 | 3.77 | 3.01 | 4.15 | 4.44 | 7.99 | **2.3** |

Table 4: **Zero-shot depth super-resolution** (e.g. low-resolution prior). ARKitScenes and RGB-D-D provide captured low-resolution depth. For other datasets, the low-resolution maps are created by downsampling the GT depths by 8 or 16 times, and the results are reported in AbsRel ↓,.

| Model | Encoder | NYUv2 | | | ScanNet | | | ETH-3D | | | DIODE | | | KITTI | | | Ave. Rank |
|---|---|---|---|---|---|---|---|---|---|---|---|---|---|---|---|---|---|
| | | Range | Shape | Object | Range | Shape | Object | Range | Shape | Object | Range | Shape | Object | Range | Shape | Object | |
| DAv2 | ViT-L | 17.40 | 5.24 | 6.56 | 16.75 | 4.64 | 6.74 | 68.76 | 8.23 | 19.22 | 51.55 | 29.20 | 13.41 | 31.12 | 14.93 | 17.94 | 6.4 |
| Depth Pro | ViT-L | 10.89 | 9.20 | 6.52 | 16.76 | 15.39 | 6.80 | **10.37** | 34.28 | 17.28 | 37.44 | 34.74 | 13.53 | **14.51** | 16.11 | 8.19 | 5.4 |
| Omni-DC | - | 23.24 | 5.94 | 13.79 | 22.89 | 5.44 | 8.71 | 29.47 | 4.81 | 17.97 | 38.83 | 7.75 | 25.43 | 35.42 | 8.94 | 15.06 | 6.6 |
| Marigold-DC | SDv2 | 19.83 | 2.37 | 6.18 | 17.14 | **1.97** | 6.66 | 25.36 | 2.15 | 7.72 | 39.33 | 7.59 | 18.97 | 33.44 | 9.21 | 7.72 | 4.7 |
| DepthLab | SDv2 | 23.85 | 2.66 | 10.87 | 21.17 | 2.08 | 10.40 | 30.61 | 2.75 | 10.53 | 41.01 | 6.51 | 17.17 | 40.43 | 13.60 | 18.66 | 6.4 |
| PromptDA | ViT-L | 36.67 | 20.88 | 23.14 | 35.86 | 17.87 | 21.89 | 46.21 | 24.94 | 27.42 | 49.50 | 25.66 | 28.29 | 55.79 | 32.74 | 38.29 | 8.7 |
| PriorDA (ours) | DAv2-B+ViT-S | 16.86 | 2.30 | 5.72 | **14.29** | 2.01 | 5.87 | 21.16 | 1.98 | 6.52 | 36.59 | 5.58 | 10.77 | 30.04 | 6.67 | 7.99 | 2.7 |
| | DAv2-B+ViT-B | 16.61 | 2.30 | **5.49** | 14.48 | 1.99 | **5.73** | 21.90 | 1.76 | **6.09** | 36.64 | 5.94 | **9.72** | 30.79 | 6.29 | 7.52 | 2.3 |
| | Depth Pro+ViT-B | 16.31 | **2.17** | 5.59 | 14.18 | 1.98 | 5.87 | 22.72 | 1.76 | 6.21 | **34.90** | 4.86 | 11.99 | 30.44 | **5.47** | 6.04 | **1.9** |

Table 5: **Zero-shot depth inpainting** (e.g. missing area prior). All results are reported in AbsRel ↓. Metrics are calculated only on the masked (inpainted) regions. "Range": masks for depth beyond 3m (indoors) and 15m (outdoors), "Shape": average result for square masks of sizes 80, 120, 160, and 200, "Object": object segmentation masks detected by YOLO (Jocher et al., 2023).

tured low-resolution depth. **To cover "any prior"**, we construct 9 individual patterns: sparse points (SfM, LiDAR, Extremely sparse), low-resolution (captured, x8, x16), and missing areas (Range, Shape, Object). We also mix these patterns to simulate more complex scenarios.

**Baselines** We compare with two kinds of methods: *1) Post-aligned MDE*: Depth Anything v2 (DAv2) (Yang et al., 2024b) and Depth Pro (Bochkovskii et al., 2025); and *2) Prior-based MDE*: Omni-DC (Zuo et al., 2024), Marigold-DC (Viola et al., 2024), DepthLab (Liu et al., 2024a) and PromptDA (Lin et al., 2025).

## 4.2 COMPARISON ON MIXED DEPTH PRIOR

We quantitatively evaluate the ability to handle challenging unseen mixed priors in Tab 2. In terms of absolute performance, all versions of our model outperform compared baselines. More importantly, our model is less impacted by the additional patterns. For example, compared to the setting that only uses sparse points in Tab 3, adding missing areas or low-resolution results in only minor performance drops (1.96→2.01, 3.08 in NYUv2). In contrast, Omni-DC (2.63→2.86, 3.81), and Marigold-DC (2.13→2.26, 3.82) show larger declines. These results highlight the robustness of our method.

## 4.3 COMPARISON ON INDIVIDUAL DEPTH PRIOR

**Zero-shot Depth Completion** Tab 3 shows the zero-shot depth completion results with different kinds and sparsity levels of sparse points as priors. Compared to Omni-DC (Zuo et al., 2024) and Marigold-DC (Viola et al., 2024), which are specifically designed for depth completion and rely on sophisticated, time-consuming structures, our approach achieves better overall performance with simpler and more efficient designs.

**Zero-shot Depth Super-resolution** In Tab 4, we present results for super-resolution depth maps. On benchmarks where low-resolution maps are created through downsampling (e.g. NYUv2 (Silberman et al., 2012), ScanNet (Dai et al., 2017), *etc*), our approach achieves performance comparable to state-of-the-art methods. However, since downsampling tends to include overly specific details from the GT depths, directly replicating noise and blurred boundaries from GT leads to better results instead. Therefore, ARKitScenes (Baruch et al., 2021) and RGB-D-D (He et al., 2021b) are more representative and practical, as they use low-power cameras to capture the low-resolution depths. On these two benchmarks, our method significantly outperform other zero-shot methods.

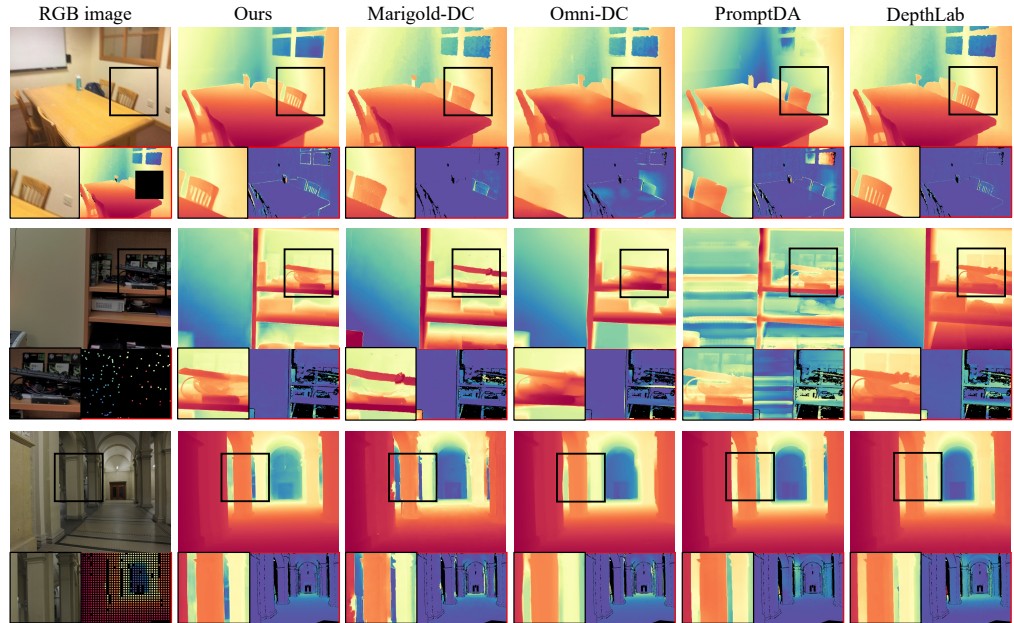

Figure 3: Qualitative comparisons with previous methods. The error map (GT vs. prediction) is shown directly below each prediction. The used depth prior is provided under the RGB image.

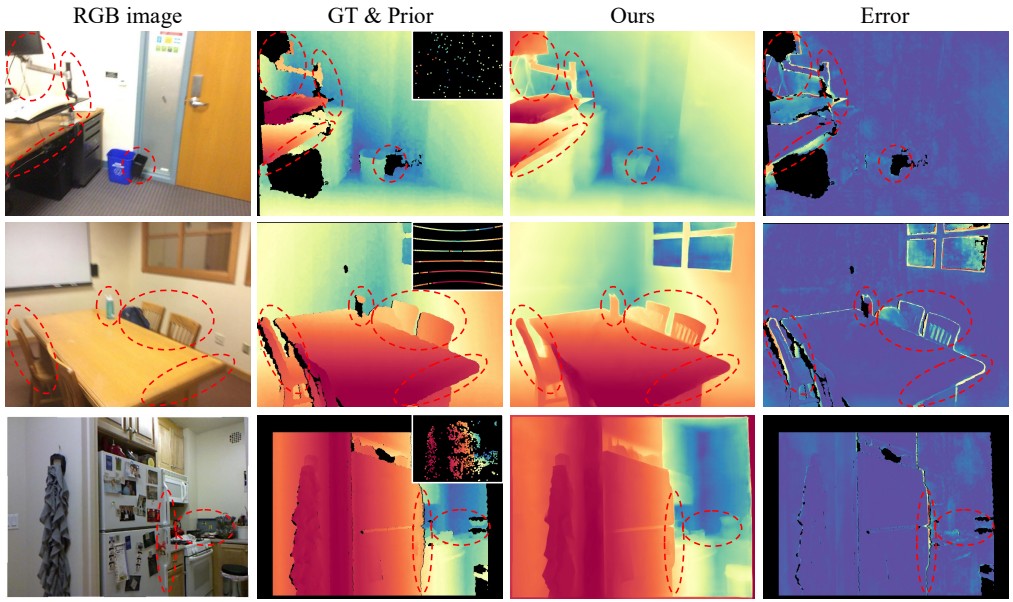

Figure 4: Error analysis on widely used but indeed noisy benchmarks (Silberman et al., 2012; Dai et al., 2017). Red means higher error, while blue indicates lower error.

**Zero-shot Depth Inpainting** In Tab 5, we evaluate the performance of inpainting missing regions in depth maps. In the practical and challenging "Range" setting, our method achieves superior results, which is highly meaningful for improving depth sensors with limited effective working ranges. Additionally, it outperforms all alternatives in filling square and object masks, demonstrating its potential for 3D content generation and editing.

## 4.4 QUALITATIVE ANALYSIS

In Fig 3, we provide a qualitative comparison of the outputs from different models. Our model consistently outperforms previous approaches, offering richer details, sharper boundaries, and more accurate metrics.

|  | S | L | M | S+M | L+M | S+L |
|---|---|---|---|---|---|---|
| Interpolation | 7.93 | 3.88 | 8.96 | 8.38 | 4.55 | 7.99 |
| Ours (w/o re-weight) | 2.92 | **3.44** | 6.91 | 3.22 | 4.53 | **4.36** |
| Ours | **2.42** | 3.51 | **6.70** | **2.60** | **4.32** | 4.40 |

Table 6: Accuracy of pre-filled depth maps with different strategies. We compare the pre-filled dense prior (i.e., $\hat{\mathbf{D}}_{\text{prior}}$) with ground truth.

| Model | Encoder | S | L | M | S+M | L | S+L |
|---|---|---|---|---|---|---|---|
| Depth Anything V2 | ViT-S | 2.15 | 2.77 | 2.68 | 2.22 | 2.87 | 3.20 |
|  | ViT-B | 1.97 | 2.73 | 2.50 | 2.02 | 2.82 | 3.09 |
|  | ViT-L | 1.92 | 2.71 | 2.29 | 1.97 | 2.79 | 3.04 |
|  | ViT-G | **1.87** | **2.70** | 2.22 | **1.94** | **2.76** | **3.02** |
| Depth Pro | ViT-L | 1.96 | 2.74 | 2.35 | 2.01 | 2.82 | 3.08 |

Table 8: Effect of using different frozen MDE models. The conditioned MDE model is ViT-B version here.

| Metric | Geometry | S | L | M | S+M | L+M | S+L |
|---|---|---|---|---|---|---|---|
| ✗ | ✓ | 5.46 | 5.29 | 5.48 | 5.36 | 5.30 | 5.46 |
| ✓ | ✗ | 2.10 | 2.94 | 2.58 | 2.17 | 3.02 | 3.31 |
| ✓ | ✓ | **1.96** | **2.74** | **2.48** | **2.01** | **2.82** | **3.08** |

Table 7: Effect of each condition for conditioned MDE models.

|  | Seen | Unseen | | | | |
|---|---|---|---|---|---|---|
|  | S | L | M | S+M | L+M | S+L |
| None | 2.50 | 3.71 | 46.07 | 2.50 | 3.74 | 3.64 |
| Interpolation | 3.40 | **2.68** | 4.28 | 3.53 | 2.94 | 3.56 |
| Ours (w/o re-weight) | 2.13 | 2.86 | 2.58 | 2.19 | 2.94 | 3.25 |
| Ours | **1.99** | 2.82 | **2.26** | **2.06** | **2.90** | **3.11** |

Table 9: Effect of pre-filled strategies on generalization. We train models with various pre-fill strategies using only sparse points and evaluate their ability to generalize to unseen types.

Fig 4 visualizes the error maps of our model. The errors mainly occur around blurred edges in the "ground truth" of real data. Our method effectively corrects the noise in labels while aligning with the metric information from the prior. These "beyond ground truth" cases highlight the potential of our approach in addressing the inherent noise in depth measurement techniques. More visualizations can be found in the supp.

## 4.5 ABLATION STUDY

We use Depth Anything V2 ViT-B as the frozen MDE and ViT-S as the conditioned MDE for ablation studies by default. All results are evaluated on NYUv2.

**Accuracy of different pre-fill strategy**    As shown in Tab 6, our pre-fill method outperforms simple interpolation across all scenarios by explicitly utilizing the precise geometric structures in depth prediction. Additionally, the re-weight mechanism further enhances performance.

**Effectiveness of fine structure refinement**    Comparing the pre-filled coarse depth maps in Tab 6 with the final output accuracy in Tab 3, 4, 5 and 2, the performance improvements after fine structure refinement (sparse points: 2.42→2.01, low-resolution: 3.51→2.79, missing areas: 6.70→2.48, S+M: 2.60→2.09, L+M: 4.32→2.88, S+L: 4.40→3.17) demonstrate its effectiveness in rectifying misaligned geometric structures in pre-filled depth maps while maintaining its metric information.

**Effectiveness of metric and geometry condition**    We evaluate the impact of metric and geometry guidance for the conditioned MDE model in Tab 7. The results show that combining both conditions achieves the best performance, emphasizing the importance of reinforcing geometric information during the fine structure refinement stage.

**Testing-time improvement**    We investigate the potential of test-time improvements in Tab 8. Our findings reveal that larger and stronger frozen MDE models consistently bring higher accuracy, while smaller models maintain competitive performance and enhance the efficiency of the entire pipeline. These findings underscore the flexibility of our model and its adaptability to various scenarios.

**Pre-fill strategy for generalization**    From Tab 9, we observe that our pixel-level metric alignment helps the model generalize to new prior patterns, and the re-weighting strategy further enhances the robustness by improving the accuracy of the pre-filled depth map.

## 5 CONCLUSION

In this work, we present ***Prior Depth Anything***, a robust and powerful solution for prior-based monocular depth estimation. We propose a coarse-to-fine pipeline to progressively integrate the metric information from incomplete depth measurements and the geometric structure from relative depth predictions. The model offers three key advantages: 1) delivering accurate and fine-grained

depth estimation with any type of depth prior; 2) offering flexibility to adapt to extensive applications through test-time module switching; and 3) showing the potential to rectify inherent noise and blurred boundaries in real depth measurements.

## ACKNOWLEDGEMENTS

This work was supported by National Natural Science Foundation of China under Grant (No.62572423, No.624B2128, No.62422606), Hong Kong Research Grant Council General Research Fund (No.17213925) and State Grid Zhejiang Electric Power Cooperation Technology Project (No.B311DS240012)

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

# A APPLICATION

| | Monocular Depth Estimation | | | Multi-view Depth Estimation | |
|---|---|---|---|---|---|
| | NYU | ETH-3D | KITTI | ETH-3D | KITTI |
| VGGT | 3.54 (-) | 4.94 (-) | 6.56 (-) | 2.46 (-) | 18.75 (-) |
| +Omni-DC | 4.12 (0.58) | 6.08 (1.14) | 6.85 (0.29) | 2.64 (0.18) | 18.66 (-0.09) |
| +Marigold-DC | 4.06 (0.52) | 5.43 (0.49) | 7.63 (1.07) | 2.81 (0.35) | 18.86 (0.11) |
| +DepthLab | 3.56 (0.02 ) | 4.92 (-0.02) | 7.97 (1.41) | 2.25 (-0.21) | 19.47 (0.72) |
| +PromptDA | **3.43 (-0.11)** | 4.97 (0.03) | 6.50 (-0.06) | 2.48 (0.02) | 18.91 (0.16) |
| +PriorDA | 3.45 (-0.09) | **4.43 (-0.51)** | **6.39 (-0.17)** | **1.99 (-0.47)** | **18.61 (-0.14)** |

Table 10: Results of refining VGGT depth prediction with different methods. All results are reported as AbsRel.

To demonstrate our model's real-world applicability, we employ prior-based monocular depth estimation models to refine the depth predictions from VGGT Wang et al. (2025a), a state-of-the-art 3D reconstruction foundation model. VGGT provides both a depth and confidence map. We take the top 30% most confident pixels as depth prior and apply different prior-based models to obtain finer depth predictions. [1]

Table 10 reports VGGT's performance in monocular and multi-view depth estimation, along with the effectiveness of different prior-based methods as refiners. We observe that only our PriorDA consistently improves VGGT's predictions, primarily due to its ability to adapt to diverse priors. These surprising results highlight PriorDA's broad application potential.

# B INFERENCE EFFICIENCY ANALYSIS

| Model | Encoder | Param | Latency(ms) |
|---|---|---|---|
| Omni-DC | - | 85M | 223 |
| Marigold-DC | SDv2 | 1,290M | 28,607 |
| DepthLab | SDv2 | 2,080M | 9,078 |
| PromptDA | ViT-L | 340M | 108 |
| PriorDA (ours) | DAv2-B+ViT-S | 97M+25M | 79 $_{49+9+21}$ |
| | DAv2-B+ViT-B | 97M+98M | 107 $_{49+9+49}$ |
| | Depth Pro+ViT-B | 952M+98M | 941 $_{883+9+49}$ |

Table 11: Analysis of inference efficiency. "$_{x+x+x}$" represents the latency of the frozen MDE model, coarse metric alignment, and conditioned MDE model, respectively.

In Tab 11, we analyze the inference efficiency of different models on one A100 GPU for an image resolution of $480\times640$. Overall, compared to previous approaches, our model variants achieve leading performance while demonstrating certain advantages in parameter number and inference latency. For a more detailed breakdown, we provide the time consumption for each stage of our method. Notably, the coarse metric alignment, which relies on kNN and least squares, introduces negligible overhead. Our method demonstrates significant efficiency advantages over previous methods.

# C MORE ABLATION STUDY

**Effect of $k$-value in Coarse Metric Alignment**  In Tab 12, we analyze the impact of the $k$-value on the accuracy of the final output $\mathbf{D}_{\text{output}}$. Overall, our method is non-sensitive to the selection of $k$-value, with most selections yielding strong structural results. This highlights the effectiveness of the nearest neighbor approach in preserving detailed metric information.

---

[1] For models less adept at handling missing pixels (DepthLab, PromptDA), the entire VGGT depth prediction was provided as prior.

|  | S | L | M | S+M | L+M | S+L |
|---|---|---|---|---|---|---|
| $k$=3 | 2.00 | 2.52 | 2.74 | 2.10 | 2.83 | **3.07** |
| $k$=5 | **1.97** | **2.16** | 2.73 | **2.04** | **2.82** | 3.09 |
| $k$=10 | 2.00 | 2.31 | 2.74 | 2.09 | 2.83 | 3.12 |
| $k$=20 | 2.10 | 2.27 | 2.76 | 2.16 | 2.83 | 3.14 |

Table 12: Impact of different $k$-value on the accuracy of the final output $\mathbf{D}_{\text{output}}$.

|  | NYU | | | | | |
|---|---|---|---|---|---|---|
|  | S | L | M | S+M | L+M | S+L |
| Omni-DC | 2.6 | 3.1 | 6.6 | 2.8 | 3.3 | 3.8 |
| PromptDA | 17.0 | **1.8** | 21.4 | 17.0 | 3.8 | 17.1 |
| PriorDA$_{\text{SP-only}}$ | 2.0 | 2.8 | **2.3** | 2.1 | 2.9 | 3.1 |
| PriorDA$_{\text{LW-only}}$ | 2.2 | 3.0 | 3.4 | 2.4 | 3.3 | 3.7 |
| PriorDA$_{\text{ALL}}$ | **2.0** | 2.7 | 2.5 | **2.0** | **2.8** | **3.1** |

Table 13: Impact of used prior patterns during training.

**Trained on Single Prior** PriorDA trained with sparse point or low-resolution prior is compared with baselines in Tab. 13, showing advantages over baseline trained with same single prior. This indicates that our method's ability to generalize to any form of prior stems from the coarse metric alignment, rather than the use of multiple patterns during training.

## D  MORE QUALITATIVE RESULTS

To further explore the boundaries of our model's capabilities and its potential to rectify the "ground truth" depth, we offer more error analysis with different patterns of priors on the 7 unseen datasets (Figure 5 for RGB-D-D, Figure 6 for ARKitScenes, Figure 7 for NYUv2, Figure 8 for ScanNet, Figure 9 for ETH-3D, Figure 10 for DIODE, Figure 11 for KITTI).

## E  MORE TRAINING DETAILS

For each training sample, we randomly select one of three patterns (e.g., sparse points, low-resolution, or missing areas) with equal probability to sample the depth prior from the ground truth depth map. Specifically: for sparse points, we randomly select 100 to 2000 pixels as valid; for low-resolution, we downsample the GT map by a factor of 8; and for missing areas, we generate a random square mask with a side length of 160 pixels. It is worth mentioning that, we find that using multiple patterns or only using sparse points lead to similar results, as shown in Tab 13. This indicates that our method's ability to generalize to any form of prior stems from the coarse metric alignment, rather than the use of multiple patterns during training.

## F  MORE DISSCUSION

**Why PromptDA preforms poor in mixed prior, sparse point and missing area?** PromptDA is only trained on low-resolution but complete priors. Therefore, it struggles to handle incomplete priors, even if we already apply DepthLab's pre-processing to fill in missing areas.

## G  USAGE OF LARGE LANGUAGE MODELS

In this work, we employed Large Language Models (LLMs) in the writing phase for sentence-level polishing.

## H    LIMITATIONS AND FUTURE WORKS

Currently, our largest conditioned MDE model is initialized with Depth Anything v2 ViT-B. Given that larger versions of the Depth Anything v2 model exhibit stronger capabilities, training conditioned MDE models based on larger backbones is an important direction for future work. Additionally, following Depth Anything, all training images are resized to 518×518. In contrast, PromptDA is natively trained at 1440×1920 resolution. Therefore, training at higher resolutions to better handle easily accessible high-resolution RGB images is another crucial direction for our future research.

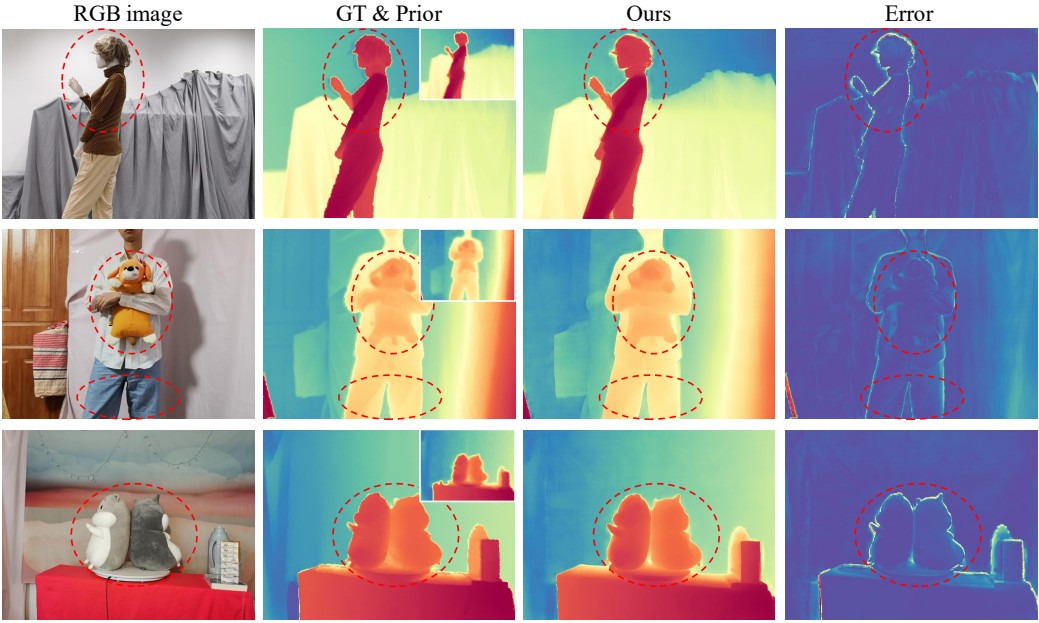

Figure 5: Error analysis on RGB-D-D.

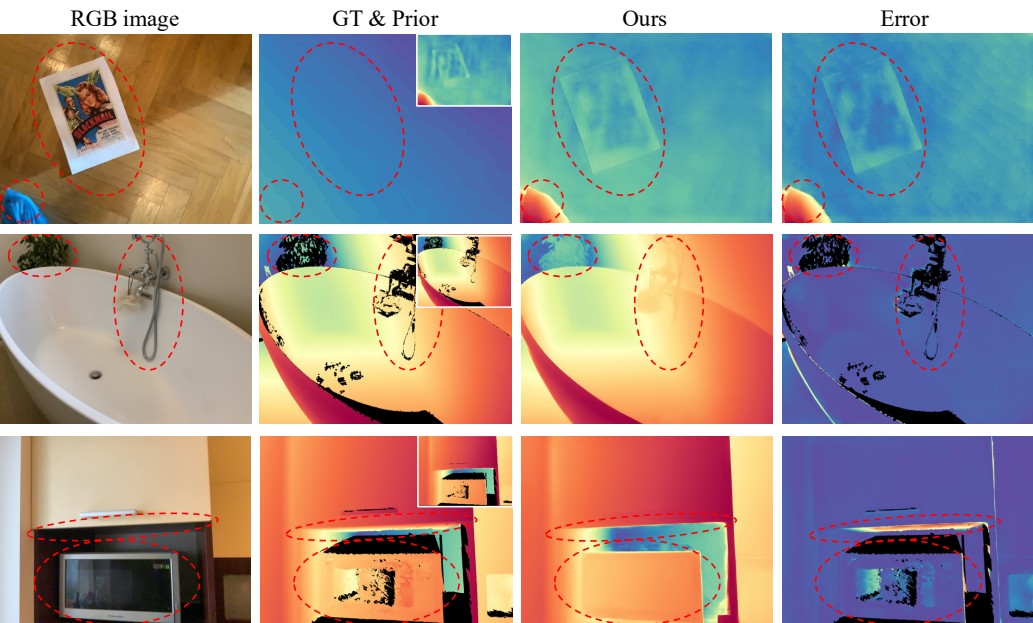

Figure 6: Error analysis on ARKitScenes.

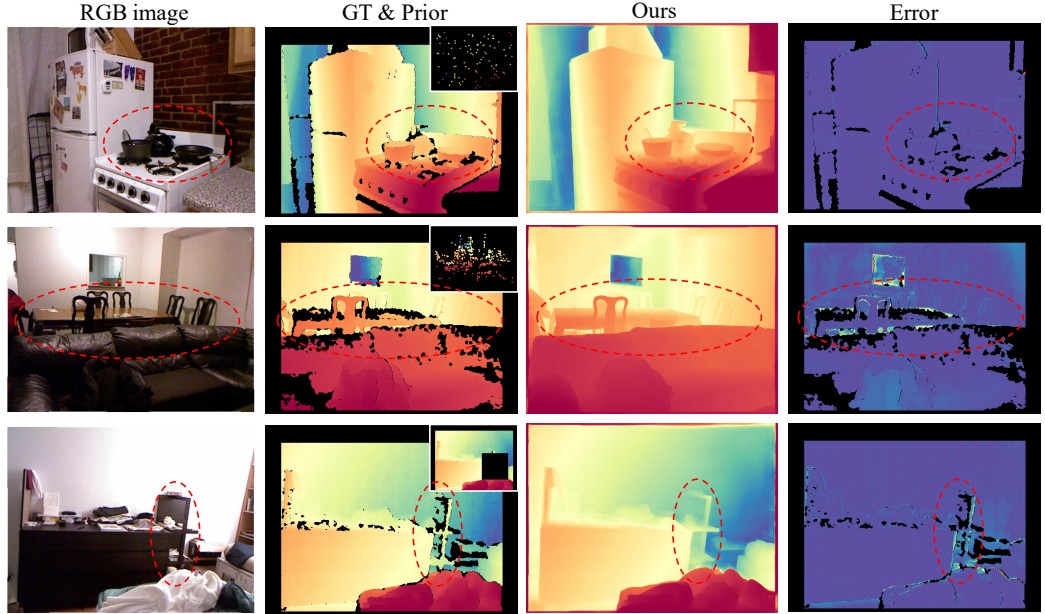

Figure 7: Error analysis on NYUv2.

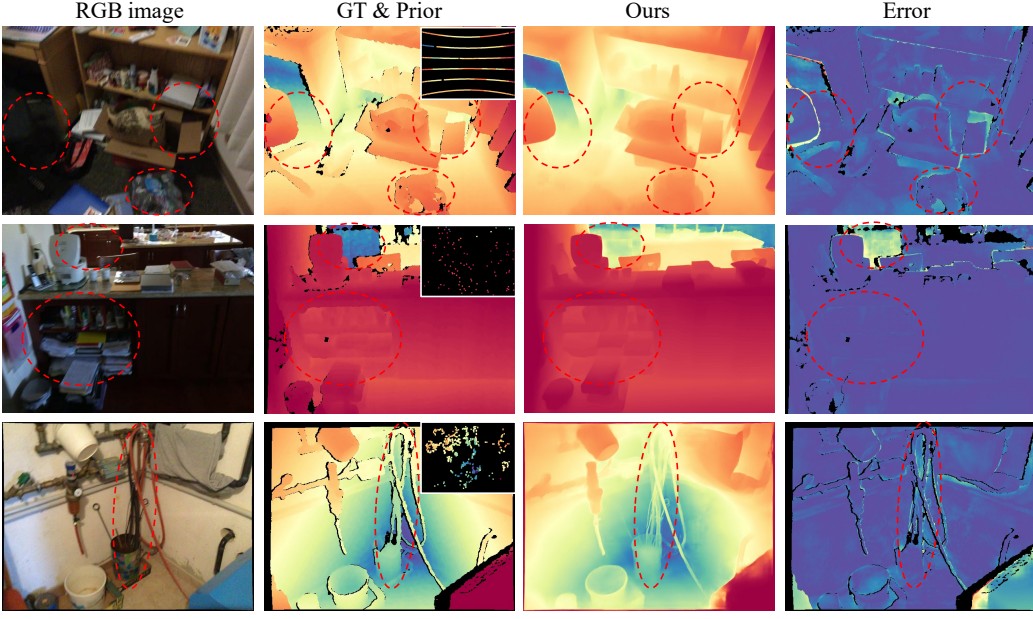

Figure 8: Error analysis on ScanNet.

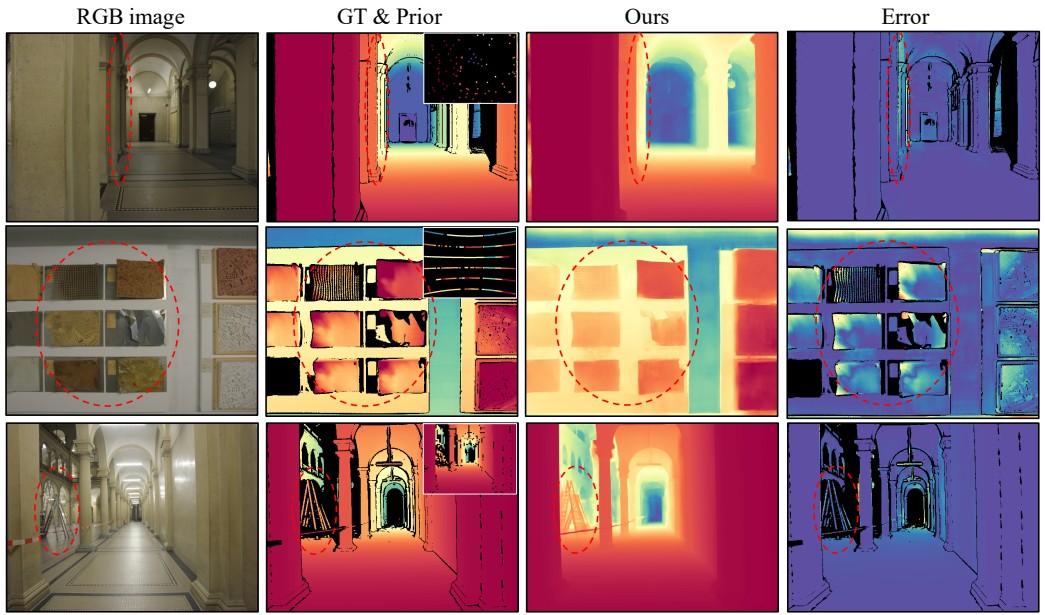

Figure 9: Error analysis on ETH-3D.

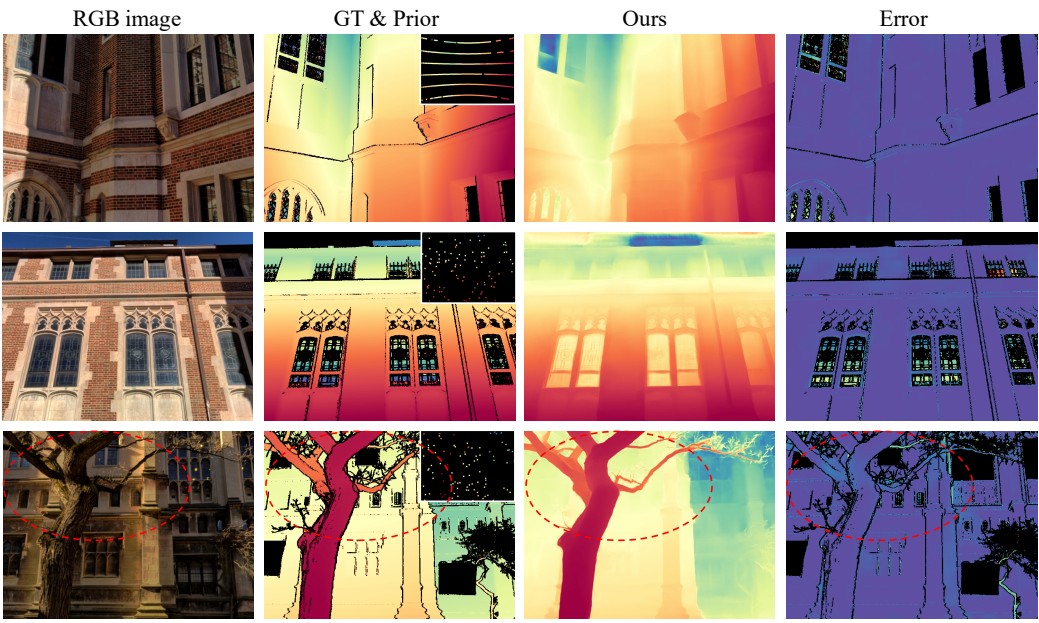

Figure 10: Error analysis on DIODE.

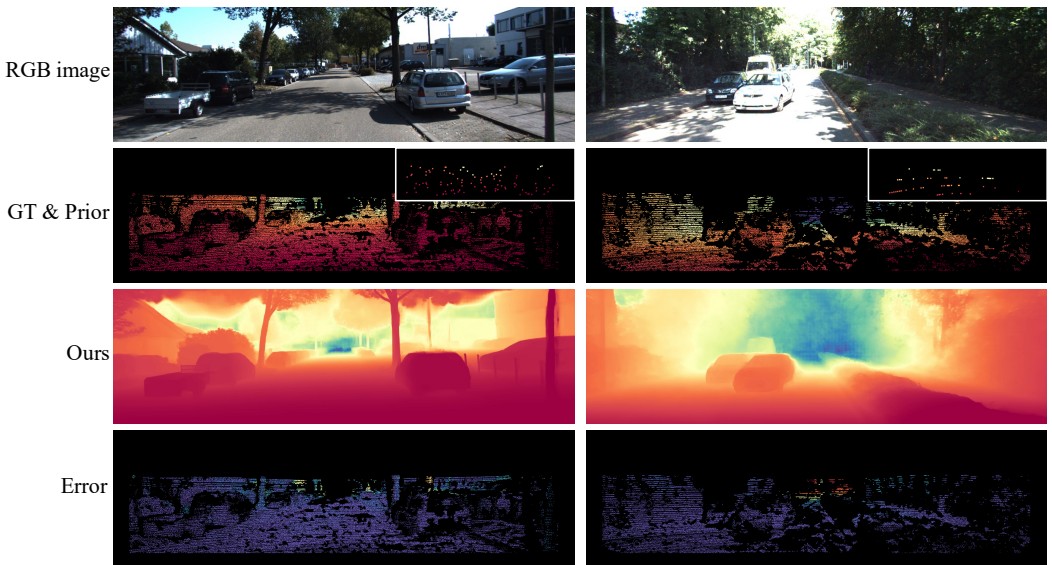

Figure 11: Error analysis on KITTI.

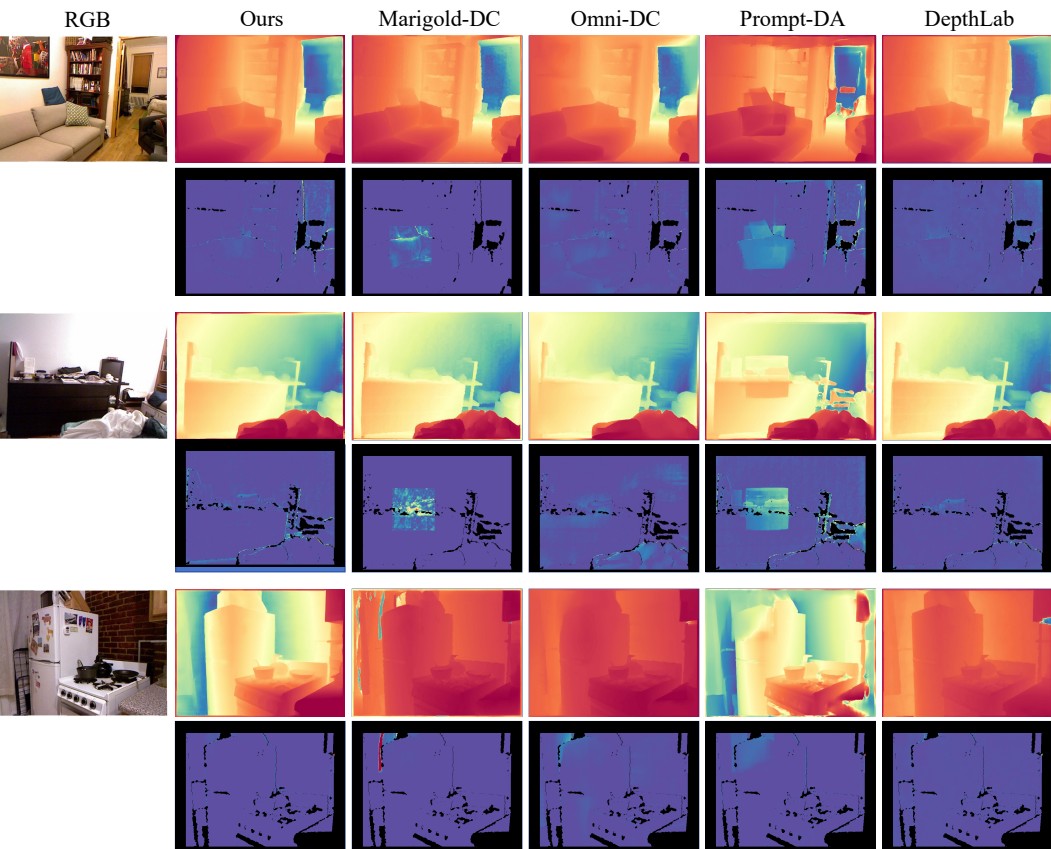

Figure 12: Qualitative comparison on NYU.

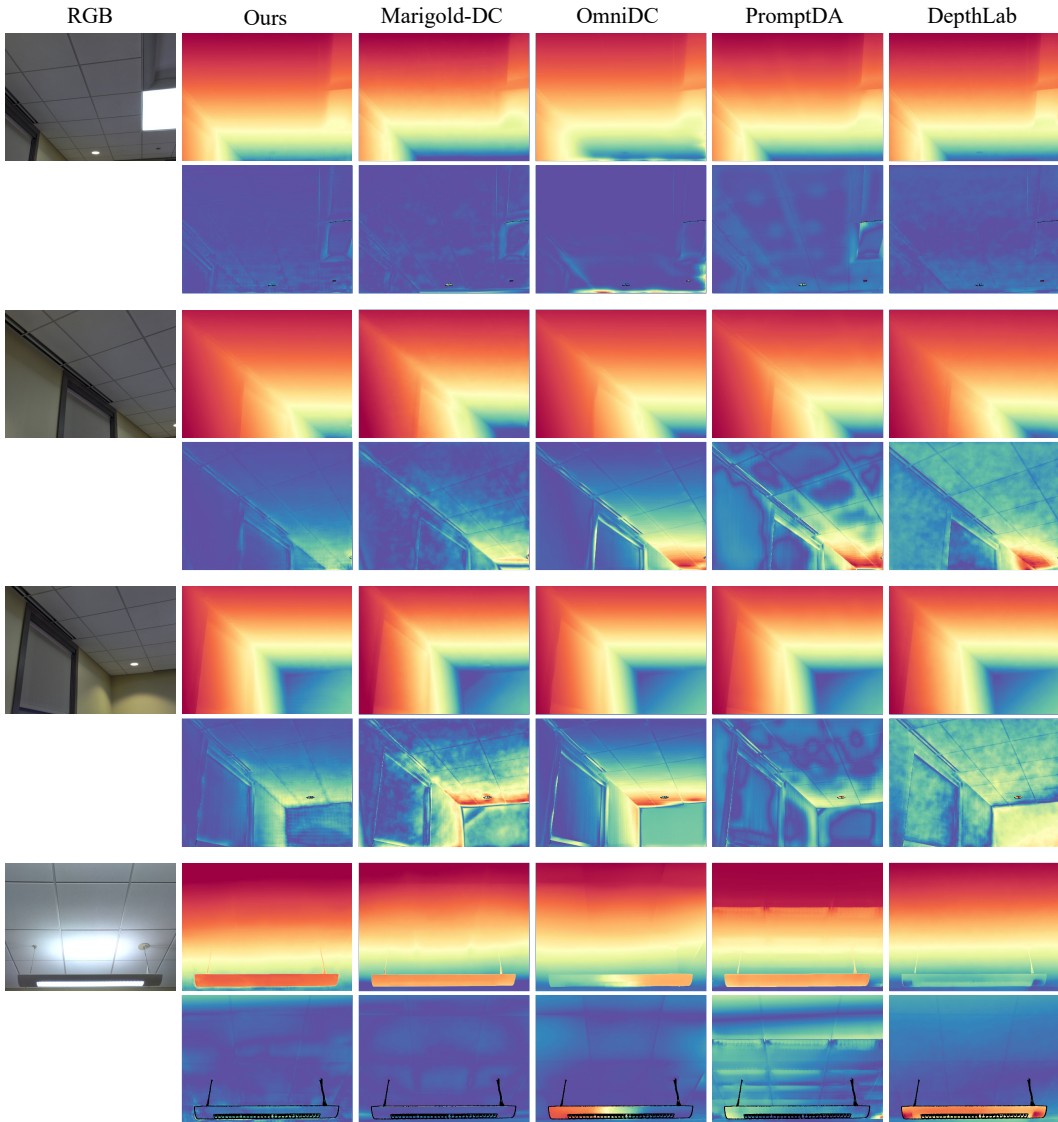

Figure 13: Qualitative comparison on DIODE.

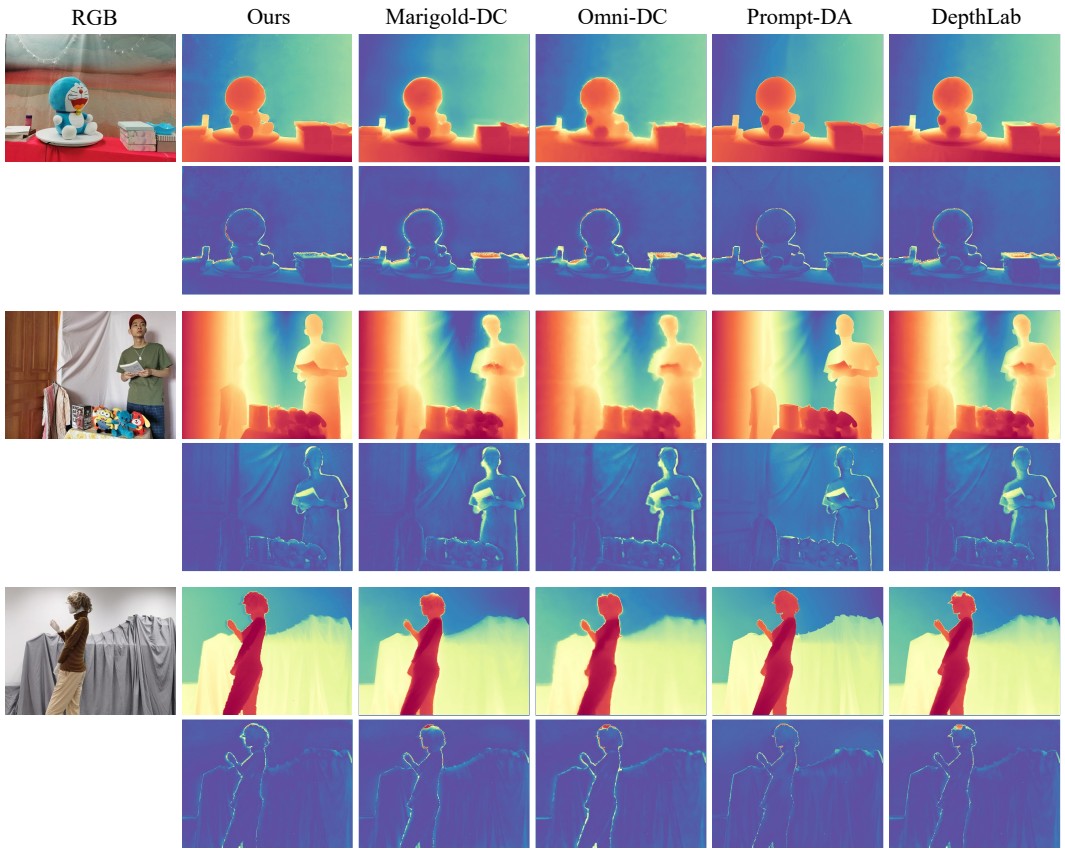

Figure 14: Qualitative comparison on RGBDD.

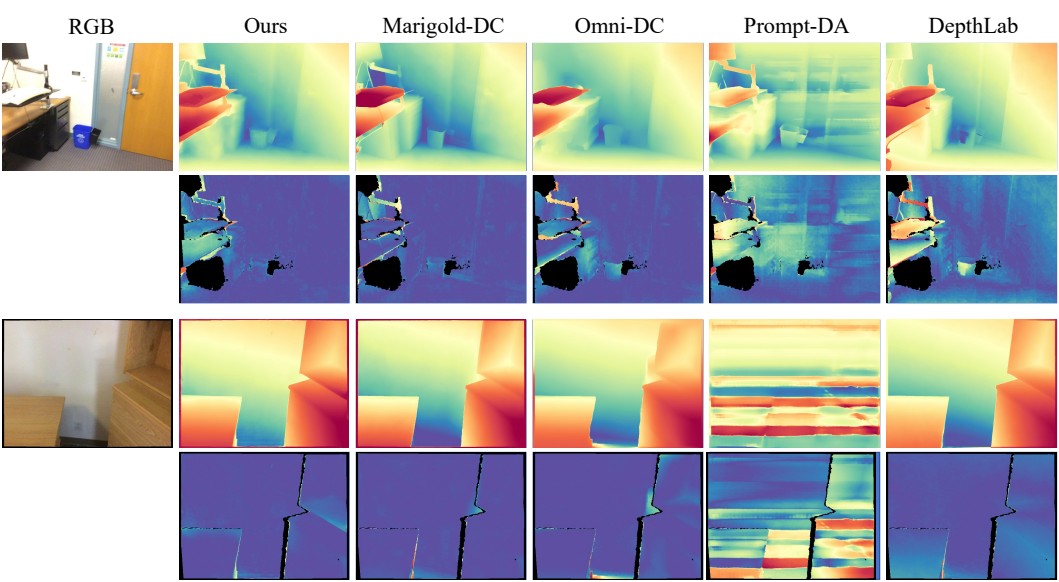

Figure 15: Qualitative comparison on ScanNet.

