# OpenReview forum: "Depth Anything with Any Prior"
_ICLR.cc/2026/Conference — ICLR 2026 Poster_

### Official Review · Reviewer_pbPR · 2025-10-29

**Soundness:** 3
**Presentation:** 4
**Contribution:** 3
**Rating:** 6
**Confidence:** 4

**Summary:**

This work proposes a metric-scale depth completion framework that can handle various types of sparse depth measurements, including LiDAR-like, SfM, masked, or range-out. The method combines affine-invariant depth estimation (referred to as geometric estimation in the paper) and metric measurement into a single pipeline to take advantage of both geometric consistency and metric accuracy. The framework is composed of two sequential stages: a monocular depth estimation (MDE) model and a conditional MDE model. The latter takes as input the RGB image, the output of the first MDE, and the filled metric measurements. Under the proposed evaluation setup, the model shows strong zero-shot performance in depth completion, super-resolution, and inpainting tasks. Ablation studies on how to fill metric measurements and how to configure the conditional MDE input further justify the design choices.

**Strengths:**

1. The proposed pipeline, which sequentially connects an MDE and a conditional MDE, may appear simple but is both reasonable and effective, as it integrates the strengths of geometric estimation and metric measurement within a unified framework.
2. Through ablation studies on pre-filled depth and the input configuration of the conditional MDE, the authors validate their design choices. The final model achieves consistently strong performance across diverse evaluation settings.

**Weaknesses:**

1. Experimental details
- The experimental section lacks sufficient detail regarding how different types of degraded depth data were generated. Specifically, it is unclear how sparse points (from SfM, LiDAR, or extremely sparse settings), low-resolution inputs (captured, ×8, ×16), and missing-area cases (range, shape, object) were obtained for each dataset. I would recommend providing explicit descriptions in the supplementary material on how these variations were constructed per dataset.
- The comparison table does not include citations for the baseline methods, and some references are missing from the main text as well. The visual emphasis in the table (e.g., bold, underline, gray shading) is inconsistent. What does “gray” indicate?

2. Figures and presentation clarity
- The visual presentation in Figures 3 and 4 is somewhat confusing. It is not entirely clear whether the error maps correspond to differences between output of proposed method and GT. It would be better to make the visualization and caption clearer to avoid misinterpretation.

3. Evaluation protocol and additional benchmarks
- While the proposed evaluation setup demonstrates strong performance, it would strengthen the paper to include additional experiments following the evaluation setups used in Marigold-DC or Zero-DC [a1], particularly on the VOID dataset. If feasible, including results on IBims would also help assess the outstanding performance of proposed method.

[a1] Hyoseok et al., "Zero-shot Depth Completion via Test-time Alignment with Affine-invariant Depth Prior", AAAI 2025.

**Questions:**

1. For the de-normalization process converting the predicted depth to metric scale, could the authors clarify whether the de-normalization process corresponds to applying the inverse of the normalization function used for metric measurements?
2. In Section 4.5 (Effectiveness of Metric and Geometry Condition and Table 7), was the training setup identical to the main experiments?
3. In Line 313, the paper mentions training with 8 GPUs. Which GPU model was used? and how long does it take?
4. In Section 4.5, is the conditional MDE trained using a single MDE backbone, or are multiple backbones used for the ablation study in Table 8?
5. There appears to be a typo in Line 455. It likely refers to Table 5 instead of Fig. 6.

**Details Of Ethics Concerns:**

This paper focuses on the depth completion task, i.e., metric-depth estimation given sparse observations (e.g., LiDAR). The authors utilize publicly available datasets. I do not identify any ethical concerns related to the proposed methodology or the datasets used in this work.

---

> ### Author Response · Authors · 2025-11-21
> **Response to Reviewer pbPR**
>
> ---
> ## W1.1: Detail about hot generating different types of depth prior.
>
> - **Sparse point**: For "SfM" and "LiDAR", we follow OmniDC's implementation. For "Extreme" sparse, we randomly selected 100 point in GT depth as prior.
>
> - **Low-resolution**: "Captured" means the benchmarks provide the low-resolution map. "x8" and "x16" mean we downscale the GT depth by 8 and 16 times.
>
> - **Missing-area**: "Range" masks depth beyond 3m (indoors) or 15m (outdoors). "Shape" uses square masks (80, 120, 160, or 200) at random positions. "Object" masks YOLO-detected object bounding boxes.
>
> We add the further detail in the captions of each table in the updated manuscript.
>
> ---
> ## W1.2: Missing citations and visual emphasis in the table.
> Thanks for your remind, We add the citations of compared baselines in the updated manuscript. The gray shading indicates that the baseline was trained on that specific dataset (i.e., the results are not "zero-shot").
>
> ---
> ## W2: Visual presentation of Figure 3 and 4.
> Yes, the error maps show the differences between the proposed method's output and the GT. We clarify the captions in the updated manuscript to prevent misinterpretation.
>
> ---
> ## W3: Results on VOID and IBims.
>
> We evaluate methods on the VOID (with provided prior) and iBim (with constructed prior patterns) benchmarks
>
> **VOID Benchmark**
>
> 150, 500, and 1,500 correspond to three density levels of depth prior provided by the VOID dataset.
> | Method              | 150   | 500   | 1500  |
> |---------------------|-------|-------|-------|
> | OMNI-DC             | 5.70  | 3.90  | **2.60**  |
> | Marigold-DC         | 5.27  | 4.31  | 3.12  |
> | PromptDA            | 14.55 | 14.10 | 13.80 |
> | DepthLab            | 9.54  | 7.50  | 5.98  |
> | Zero-DC             | 19.13 | 13.12 | 7.07  |
> | PriorDA(vitb+vitb)  | **4.39**  |**3.59**  | 2.85  |
>
> **iBims Benchmark**
>
> "S": "Extreme" setting in sparse points, "L": "×16" in low-Resolution, "M": "Shape" (square masks of 160) in missing area.
> | Method              | S   | L | M | S+M | L+M | S+L |
> |---------------------|-------|-------------|----------|--------------|----------------------|-----------------|
> | OMNI-DC             | 2.36  | 3.49        | 4.96     | 2.45         | 3.52                 | 3.75            |
> | Marigold-DC         | 2.50  | 4.12        | 2.96     | 2.46         | 4.14                 | 4.60            |
> | PromptDA            | 17.63 | **2.02**        | 24.27    | 18.05        | 2.62                 | 18.08           |
> | DepthLab            | 5.81  | 3.21        | 2.33     | 5.36         | 3.34                 | 5.58            |
> | priorda(vitb+vitb)  | **1.88**  | 2.89        | **2.03**     | **1.85**         | **2.94**                 | **3.05**            |
>
> The results further demonstrate the robustness and advantage of our method in generalizing across varied and complex prior scenarios.
>
>
> ---
> ## Q1: De-normalization process
> Yes, the de-normalization process corresponds to the inverse of the normalization function used for metric measurements.
>
> ---
> ## Q2: Training setup for Table 7.
> Yes, all the training hyperparameter are identical to the main experiments.
>
> ---
> ## Q3: Training resource.
> Our model was trained on 8 A100 GPUs for 32 hours.
>
> ---
> ## Q4: MDE backbone used in training:
> In training, we only use the Depth Anything v2 ViT-S version as forzen MDE for efficiency.
>
> ---
> ## Q5: Typo:
> Thanks for your reminder. We corrected the typo in the updated manuscript.

---

> ### Comment · Reviewer_pbPR · 2025-11-25
>
> Thank you for your response.
> Many of my concerns have been addressed in your clarification.
> I encourage you to include additional results, such as the VOID experiments, and the points raised in my questions in the revision, as they would further improve the clarity of the paper.
> These questions were intended to clarify details that make it difficult to understand the current version paper.

---

> > ### Author Response · Authors · 2025-12-01
> > **Further Response to Reviewer pbPR**
> >
> > Thank you for your support! We will include the additional results in the subsequent version of the manuscript.

---

### Official Review · Reviewer_uZ3K · 2025-10-30

**Soundness:** 3
**Presentation:** 3
**Contribution:** 3
**Rating:** 6
**Confidence:** 4

**Summary:**

This paper proposes Prior Depth Anything, a unified framework that fuses precise but incomplete metric depth priors (e.g., sparse points, low-resolution maps, masked regions) with complete but relative monocular depth predictions to produce dense and accurate metric depth. The pipeline is coarse-to-fine:(i) Pixel-wise scale–shift alignment with distance-aware weighting fills missing regions using a frozen monocular predictor, reducing gaps among different prior types;(ii) A conditioned monocular depth network refines results from the RGB image, the pre-filled prior, and the prediction (normalized), mitigating measurement noise and reconciling geometry.Experiments on seven datasets show strong zero-shot results in completion, super-resolution, and inpainting, with robustness to mixed priors. Test-time backbone swapping supports accuracy–efficiency trade-offs. Overall, the design provides a clear fusion recipe and broad, competitive performance across settings.

**Strengths:**

Broad applicability and novelty: One framework handles completion, upsampling, inpainting, and their combinations, covering common real-world inputs that previous methods often treat separately.

Strong empirical performance: Consistent zero-shot results across multiple datasets and tasks, often matching or surpassing task-specific baselines without per-task fine-tuning.

Robust to mixed priors: Maintains accuracy when prior types co-occur (e.g., sparse + low-res + holes), a challenging but practical scenario.

Clear integration pipeline: Pixel-level metric alignment plus a conditioned refiner is simple, well-motivated, and effective in practice; normalization enables flexible backbone choices.

**Weaknesses:**

Efficiency underreported: The two-stage design (predictor + kNN-style alignment + conditioned refiner) lacks detailed latency, memory, and component-wise cost, leaving deployability unclear.

Ablations and sensitivity limited: The impact of weaker/faster predictors, distance-aware weighting, neighborhood size, or removing coarse alignment is not fully quantified.

**Questions:**

Effect of predictor quality: How do metrics change with lighter/faster monocular predictors versus the default, and with different alignment settings? Please provide an accuracy–throughput curve and key ablations.

---

> ### Author Response · Authors · 2025-11-21
> **Response to Reviewer uZ3K**
>
> ---
> ## W1: Efficiency underreported: detailed latency, memory, and component-wise cost.
> ### **For component-wise latency:**
>
> The Table 11 of the appendix provide the latency of the each component of our two-stage pipeline. We put it here:
>
> | Model               | Encoder            | Param         | Latency (ms)       |
> |---------------------|--------------------|---------------|--------------------|
> | Omni-DC             | -                  | 85M           | 223                |
> | Marigold-DC         | SDv2               | 1,290M        | 28,607             |
> | DepthLab            | SDv2               | 2,080M        | 9,078              |
> | PromptDA            | ViT-L              | 340M          | 108                |
> | PriorDA      | DAV2-B+ViT-S       | 97M+25M       | 79 (49+9+21)       |
> | PriorDA      | DAV2-B+ViT-B       | 97M+98M       | 107 (49+9+49)      |
> | PriorDA      | Depth Pro+ViT-B    | 952M+98M      | 941 (883+9+49)     |
>
> Note: The "x+x+x" in the Latency column for PriorDA represents the latency of the frozen MDE model, the coarse metric alignment, and the conditioned MDE model, respectively.
>
> In summary, our method demonstrates a significant advantage in terms of latency compared to competitive baselines. The inference time for the different variants of the Frozen MDE and Conditioned MDE variants essentially matches that of their corresponding ViT model sizes, and the kNN-based metric alignment component introduces only a minimal overhead.
>
> ### **For Memory Usage:**
>
> For the memory-intensive variant (ViT-Large + ViT-Base), the peak memory consumption is only 10.4 GB, which is highly conducive to practical deployment on common GPU platforms.
>
> ---
> ## W2&Q1: More ablations:
> We provide the accuracy–throughput of different variants with different predictors on NYU as following:
>
>
> | Predictors | S | | L | | M | | S+M | | L+M | | S+L | |
> |-----------------|----------|--------|----------|--------|----------|--------|----------|--------|----------|--------|----------|--------|
> | | AbsRel↓ | FPS↑ | AbsRel↓ | FPS↑ | AbsRel↓ | FPS↑ | AbsRel↓ | FPS↑ | AbsRel↓ | FPS↑ | AbsRel↓ | FPS↑ |
> | DAv2-vits | 2.15 | 11.9 | 2.77 | 11.7 | 2.68 | 9.8 | 2.22 | 12.5 | 2.87 | 12.0 | 3.20 | 12.7 |
> | DAv2-vitb | 1.97 | 8.7 | 2.73 | 8.6 | 2.50 | 7.5 | 2.02 | 9.1 | 2.82 | 8.9 | 3.09 | 9.1 |
> | DAv2-vitl | 1.92 | 4.7 | 2.71 | 4.6 | 2.29 | 4.2 | 1.97 | 4.7 | 2.79 | 4.5 | 3.04 | 4.7 |
> | DAv2-vitg | **1.87** | 1.7 | **2.70** | 1.7 | **2.22** | 1.6 | **1.94** | 1.7 | **2.76** | 1.7 | **3.02** | 1.7 |
> | Depth-pro-vitl | 1.96 | 0.7 | 2.74 | 0.7 | 2.35 | 0.7 | 2.01 | 0.7 | 2.82 | 0.7 | 3.08 | 0.7 |
>
> In our perspective, some of the components you mentioned are already ablated. For clarity, we summarize their locations below:
>
> - **Weaker/Faster Predictors**: This is quantified in Table 8, where we test different variants of the Depth Anything v2 (ViT-Small being the lightest/fastest predictor we found).
>
> - **Distance-Aware Weighting**: This ablation is covered in Table 6 (second and third rows), showing the effect of using or omitting the distance-aware weighting.
>
> - **Different Neighborhood Size**: The sensitivity to the size of the neighborhood is detailed in Table 12 (Appendix).
>
> - **Removing Coarse Alignment**: The effect of removing the initial coarse alignment step is shown in Table 7 (first row).
>
> Could you kindly specify if there are any additional ablation studies or components, not explicitly covered in the points above, that you would like us to investigate further?

---

> > ### Comment · Reviewer_uZ3K · 2025-11-24
> > **Official Comment by Reviewer uZ3K**
> >
> > The authors have addressed my concerns and I maintain my initial positive rating.

---

> > > ### Author Response · Authors · 2025-12-01
> > > **Further Response to Reviewer uZ3K**
> > >
> > > Thank you for your support!

---

### Official Review · Reviewer_thPV · 2025-10-30

**Soundness:** 2
**Presentation:** 2
**Contribution:** 1
**Rating:** 2
**Confidence:** 4

**Summary:**

This paper presents a framework that integrates incomplete but accurate metric depth measurements with complete yet relative geometric predictions from monocular depth estimation (MDE) models to generate detailed and metrically consistent depth maps. The motivation stems from the complementary nature of these inputs: MDE produces dense, fine-grained relative depth but lacks absolute scale, whereas metric measurements provide precise scaling information but are often sparse, low-resolution, or partially missing. The proposed method adopts a coarse-to-fine pipeline that first performs pixel-level metric alignment, followed by a conditioned MDE refinement stage. The framework demonstrates strong zero-shot generalization across depth completion, super-resolution, and inpainting tasks on seven real-world datasets.

**Strengths:**

1. The paper presents extensive experiments that effectively validate the proposed method and justify its performance against benchmark approaches.
2. The proposed framework achieves enhanced depth estimation quality through a simple and well-structured pipeline.

**Weaknesses:**

1. The proposed method shows limited novelty, as it primarily predicts per-pixel scale and shift values in MDE to align with the depth prior.
2. The paper states, “We highlight best and second-best results” in the quantitative results section; however, only a few columns in Tables 2–5 are correctly annotated. The remaining columns contain inconsistencies, including missing annotations, unclear labeling, or incorrect markings.
3. The reported quantitative and qualitative results do not convincingly demonstrate the proposed method’s superiority over baseline approaches. Specifically, the method fails to outperform in:
    + 14 out of 16 experiments in Table 4,
    + 4 out of 16 experiments in Table 5,
    + 5 out of 16 experiments in Table 3, and
    + 1 out of 3 comparisons in Figure 3 (the second experiment).

These results indicate that the proposed method underperforms in a significant portion of the evaluations.
Overall, the improvements reported do not sufficiently support the claimed performance gains.

**Questions:**

1. What specific problem does the proposed distance-aware weighting (line 229) aim to solve? The authors are encouraged to provide clearer intuition or theoretical justification for this formulation.
2. Why are recent baselines such as SharpDepth [1] not included in the comparison?
3. How were the baseline methods implemented, particularly with respect to the use of different types of depth priors? Additional implementation details would improve clarity and reproducibility.
4. It would strengthen the paper to include more qualitative comparisons with other benchmarks—ideally 10–20 diverse examples—to better illustrate the method’s generalization ability and visual performance.

[1] Pham, D.-H., Do, T., Nguyen, P., Hua, B.-S., Nguyen, K., & Rang, N. (2024). SharpDepth: Sharpening Metric Depth Predictions Using Diffusion Distillation. arXiv preprint arXiv:2411.18229.

---

> ### Author Response · Authors · 2025-11-21
> **Response to Reviewer thPV (1/3)**
>
> ---
> ## W1: Limited Novelty (The method primarily predicts per-pixel scale and shift values)
>
> Actually, our method **"do not predicts"** the per-pixel scale and shift. Our method is two-stage pipelien: First, we **"calculate"** the per-pixel scale and shift values using depth predictions with fine geometric structure to align the metric and densify the incomplete prior. Second, we employ a conditional MDE to refine the pre-filled depth map with multiple input conditions.
>
> As for detailed novelty,
>
> - **Motivation**: As we known, we are the first to focus on universally refining different patterns of depth priors, which is highly practical.
>
> - **Methodology**: We believe our two-stage pipeline provide enough techinical contribution. The first stage effectively bridges the domain gap between priors with different patterns, and the second stage fully capitalizes on the implicit geometric structure knowledge in pre-trained MDE models.
>
> - **Experiments**: We are the first to provide a unified evaluation of model generalization across various prior patterns, which is curical for real-world application. We also provide some interesting results, like refine the "ground-truth" depth map in current dataset.
>
> In summary, our work provides meaningful innovations from problem definition, method design, to experimental validation. We believe the focus on "any prior" will offer new insights to the community.
>
> ---
> ## W2: Tables are not correctly annotated.
> We apologize for the oversight. The correct best and second-best annotations have been added to the updated manuscript.

---

> ### Author Response · Authors · 2025-11-21
> **Response to Reviewer thPV (2/3)**
>
> ---
> ## W3: Bad Performance (The proposed method underperforms in a significant portion of the evaluations)
>
> ### **Statistical Analysis of Quantitative Performance:**
>
> To analyze whether our method "*underperforms in a significant portion of the evaluations,*" we calculate the average ranking of all models across the four primary evaluation tables (Tables 2-5). The results are summarized below:
>
> | Method                 | Ave. Rank (Table 2) | Ave. Rank (Table 3) | Ave. Rank (Table 4) | Ave. Rank (Table 5) | Ave. Rank (All Tables) |
> |------------------------|-------------------|-------------------|-------------------|-------------------|-----------------------|
> | DAv2                   | 7.5               | 7.3               | 8.9               | 6.4               | 7.5                   |
> | Depth Pro              | 6.7               | 6.1               | 7.8               | 5.4               | 6.5                   |
> | Omni-DC                | 4.2               | 3.7               | 3.9               | 6.6               | 4.6                   |
> | Marigold-DC            | 5.1               | 4.3               | 6.2               | 4.7               | 5.0                   |
> | DepthLab               | 7.1               | 7.6               | 6.4               | 6.4               | 6.9                   |
> | PromptDA               | 8.4               | 8.8               | 2.6               | 8.7               | 7.2                   |
> | PriorDA (DAv2-B+ViT-S) | 2.9               | 3.3               | 3.9               | 2.7               | 3.2                   |
> | PriorDA (DAv2-B+ViT-B) | 2.0               | 2.2               | 2.9               | 2.3               | 2.3                   |
> | PriorDA (Depth Pro+ViT-B) | **1.1**            | **1.7**               | **2.3**               | **1.9**               | **1.7**                   |
>
> This statistical overview demonstrates that our methods achieve the best average ranking across all settings. **Our approach appears to outperform in a significant portion of the evaluations, rather than underperforming.**
>
> Moreover, we want to emphasize that our method is designed for the unified processing of various prior patterns, making Table 2—which shows our superior performance across these diverse priors—the most crucial set of experiments. Our experiments in Table 2 show a clear and comprehensive advantage.
>
> ### **Regarding Qualitative Results (Figure 3):**
>
> Could the reviewer please more specifically indicate why our method is considered to underperform in the second experiment of Figure 3?
>
> From our perspective, the error maps of the other methods show clearly highlighted regions, indicating substantially larger errors in these areas. In contrast, PriorDA's error map shows significantly smaller, more dispersed errors, suggesting higher overall accuracy.
>
>
> ### **More Comparison on more benchmark:**
> To further demonstrate the robustness and generalization of our unified approach, we evaluate PriorDA on two additional external benchmarks: VOID and iBim.
>
> **VOID Benchmark:**
>
> 150, 500, and 1,500 correspond to three density levels of depth prior provided by the VOID dataset.
> | Method              | 150   | 500   | 1500  |
> |---------------------|-------|-------|-------|
> | OMNI-DC             | 5.70  | 3.90  | **2.60**  |
> | Marigold-DC         | 5.27  | 4.31  | 3.12  |
> | PromptDA            | 14.55 | 14.10 | 13.80 |
> | DepthLab            | 9.54  | 7.50  | 5.98  |
> | Zero-DC             | 19.13 | 13.12 | 7.07  |
> | PriorDA(vitb+vitb)  | **4.39**  |**3.59**  | 2.85  |
>
> **iBims Benchmark:**
>
> "S": "Extreme" setting in sparse points, "L": "×16" in low-resolution, "M": "Shape" (square masks of 160) in missing area.
> | Method              | S   | L | M | S+M | L+M | S+L |
> |---------------------|-------|-------------|----------|--------------|----------------------|-----------------|
> | OMNI-DC             | 2.36  | 3.49        | 4.96     | 2.45         | 3.52                 | 3.75            |
> | Marigold-DC         | 2.50  | 4.12        | 2.96     | 2.46         | 4.14                 | 4.60            |
> | PromptDA            | 17.63 | **2.02**        | 24.27    | 18.05        | 2.62                 | 18.08           |
> | DepthLab            | 5.81  | 3.21        | 2.33     | 5.36         | 3.34                 | 5.58            |
> | priorda(vitb+vitb)  | **1.88**  | 2.89        | **2.03**     | **1.85**         | **2.94**                 | **3.05**            |
>
> The results further demonstrate the robustness and advantage of our method in generalizing across varied and unseen prior scenarios.

---

> > ### Author Response · Authors · 2025-11-21
> > **Response to Reviewer thPV (3/3)**
> >
> > ---
> > ## Q1: The specific problems the distance-aware weighting aim to solve.
> > We apologize for the missing statement. The distance-aware weighting is designed to solve two specific problems:
> >
> > - Discontinuity Risk: Adjacent pixels in missing regions might select different k-nearest neighbors, leading to abrupt and unrealistic depth changes at the pixel boundary.
> >
> > - Uniform Weighting Bias: While nearby supporting points offer more reliable metric cues than distant ones, equal weighting in the least squares plane fitting overlooks this geometric correlation, leading to suboptimal local alignment.
> >
> > We also update this statement in Section 3.1 of the updated manuscript.
> >
> > ---
> > ## Q2: Why SharpDepth not included.
> > SharpDepth is designed to refine the structure of a **completed** metric depth prediction. Our task, conversely, focuses on the refinement of an **incomplete** metric depth measurement. SharpDepth's methodology does not allow it to process incomplete depth maps as input, making it unsuitable for direct comparison in our setting.
> >
> > ---
> > ## Q3: Additional implementation details about baseline.
> > - Omni-DC and Marigold-DC: We directly input the depth prior following the setup provided in their original codebases.
> >
> > - DepthLab and PromptDA: Following their implementations, we use bilinear interpolation to pre-fill the incomplete prior area and increase the image resolution to match the RGB input.
> >
> > ---
> > ## Q4: More qualitative comparisons.
> > We include more qualitative comparisons with other benchmarks in the updated manuscript. Please refer to Figure 12, 13, 14 and 15 for these additional results, which further illustrate our method's generalization ability.

---

> ### Comment · Reviewer_thPV · 2025-11-28
>
> Thank you for your detailed response and for clarifying the performance results in Tables 3, 4, and 5. Regarding the novelty concern, my point is that the approach of predicting or recalculating shift and scale is very trivial: calibrating a few pixels with absolute depth, hopefully, the adjustment generalizes to other pixels. Can you explain in detail what is the technical contributions on doing that except "calibration" stuff?

---

> > ### Author Response · Authors · 2025-12-01
> > **Further Response to Reviewer thPV**
> >
> > Thanks for your prompt feedback!
> >
> > ### For Performance Concern:
> > ---
> >
> > We appreciate that you recognized our new results. We believe these statistical results, which more clearly show the advantages of our method, can eliminate your concern regarding the comparison performance.
> >
> > ### For Novelty Concern:
> > ---
> >
> > First, we would like to further clarify that Stage 1 (COARSE METRIC ALIGNMENT), which involves the "scale and shift" calculation, **is not a "calibration" or "adjustment" step.**
> >
> > As highlighted in lines 203-204, the scale and shift calculation is primarily proposed to pre-fill missing pixels in the depth prior and consequently reduce the distribution differences caused by the varying incompleteness of different prior types. This pre-filling process significantly improves the generalizability of the conditional MDE model across diverse prior inputs. This is a well-motivated and non-trivial idea, as few existing methods uniformly handle various forms of depth prior.
> >
> > Stage 2 (FINE STRUCTURE REFINEMENT), on the other hand, mainly serves to calibrate potential noise within the pre-filled depth map. This stage leverages the pre-trained MDE's understanding of scene geometric structures to re-predict a "structurally more accurate" metric depth map based on the pre-filled input. And the Stage 2 does not involve predicting scale and shift from the original image.

---

### Official Review · Reviewer_Zgrm · 2025-11-05

**Soundness:** 4
**Presentation:** 4
**Contribution:** 3
**Rating:** 8
**Confidence:** 4

**Summary:**

The paper introduces a framework for prior-based monocular depth estimation, which accepts RGB images and depth priors (e.g. LiDAR or SfM in depth completion, low-resolution Time of Flight camera depth maps, incomplete depth maps, etc.) and outputs a dense metric depth map. Instead of focusing on any particular prior, this work proposes a unified method that can generate metric depth from any of these prior sources. The method consists of two stages. The first generates a coarse depth map based on depth priors and the predictions from a pre-trained monocular depth model (e.g. Depth Anything V2 is used in training experiments). This is done with the proposed "pixel-level metric alignment", where for each missing pixel, a set of $k$ nearest neighbors present in the prior are computed. These set of points are then used to compute a scale and shift parameter which best aligns (weighted by distance to the query pixel) these points with that predicted by the frozen MDE. The scale and shift parameters used to compute the missing pixel's values from the frozen MDE prediction. The second stage aims to improve robustness against noise in priors using another conditioned MDE. The input condition to this MDE is generated from RGB input, normalized frozen MDE prediction, and the normalized coarse prediction from the first stage via convolution layers. The output is then de-normalized to obtain the final predictions. The model is trained with synthetic datasets using several augmentations / corruptions to generate synthetic priors, and then evaluated zero-shot across multiple different real-world tasks with different sources of priors, across multiple datasets, and shown overall to perform comparably or better than existing methods.

**Strengths:**

- Extensive experiments are conducted to show that the method performs strongly across different sources of depth priors, which is shown to be a key strength of the model compared to existing works. The proposed method convincingly generalizes significantly better compared to existing state-of-the-art approaches.

- The pixel-level alignment method for in-painting missing depth regions seems novel, and shown to be significantly better than naive interpolation in Table 6 and 9. There is reason to believe that this insight can be generalized to improve other works in the field.

- Extensive ablations are also performed, ablating the different input conditions for the conditioned MDE model (Table 7), different frozen MDE models (Table 8), and in-painting strategy (Table 6 and 9).

**Weaknesses:**

- The method first requires using a heuristic-based approach for densifying a depth prior. This might not be effective or practical especially when the prior map is extremely sparse, from both a latency (since densification requires solving a least-square regression for each missing pixel) and performance (the nearest neighbors might be extremely far away from the query pixel) standpoint.

- Minor: I am not sure whether it is an issue with my PDF viewer, but the formatting of the paper seems off, especially in the first page. The hyperlinks seem to jump all over the place, and overlays / occludes existing text.

**Questions:**

- The authors provided an intuition for the two-stage approach, where the first seems to provide a coarse (possibly noisy) estimate, and the second refines this. How strongly is this observed is this in practice? In the extreme case, how robust is the method given a very noisy (or perhaps even randomly initialized) depth prior map, would it be possible to at least recover the performance of that from the pre-trained MDEs?

- The authors show that it is possible to stack multiple (2) MDEs to produce better depth predictions. I wonder if this is possible to improve results further at test-time by applying the conditional MDE in a recursive manner (i.e. repeatedly feeding in the output of the conditioned MDE in place of $\hat{D}_{prior}$ back into itself), which can allow using "adaptive" inference-time compute to produce outputs at various levels of refinement.

---

> ### Author Response · Authors · 2025-11-21
> **Response to Reviewer Zgrm**
>
> ---
> ## W1: Densifying extremely sparse depth priors (Latency and Performance)
>
> ### For latency:
>
> Due to highly optimized operators, our pre-fill process (densification) is highly parallel. We find that sparser depth prior even lead to lower latency (likely because the sorting for $k$-Nearest Neighbors becomes simpler). The densification step consistently introduces minimal overhead.
>
> On a $480 \times 640$ image, we measure the latency of pre-fill process at a single 2080ti GPU with different numbers of valid prior point:
>
>  Valid Points | Latency(ms) |
> |:-------:|:------:|
> |   5     | 9  |
> |  10     | 9  |
> |  20     | 9  |
> |  50     | 9  |
> | 100     | 10  |
> |1000     | 13  |
> |2000     | 17  |
>
> Besides, the full pipeline latency analyzed in Table 11 of our manuscript, which demonstrates that the pre-fill step introduces minimal overhead.
>
> ### For performance:
>
> The "nearest neighbors might be extremely far away from the query pixel" challenge is a fundamental difficulty for *all depth completion methods* under extremely sparse conditions.
>
> In the face of this shared challenge, our method significantly outperforms existing baselines. We provide comparative results on the NYU and ScanNet datasets using even fewer random point as depth prior.
>
> **Performance Comparison on NYU (Abs Rel, ↓)**
> |        Setting        |   5  |  10  |  20  |  50  |
> |:--------------------:|:----:|:----:|:----:|:----:|
> | OMNI-DC              |16.31 |11.31 | 7.30 | 4.03 |
> | Marigold-DC          |12.66 | 6.75 | 4.23 | 2.80 |
> | PromptDA             |34.80 |24.04 |19.26 |17.18 |
> | DepthLab             |24.20 |15.97 |10.77 | 7.72 |
> | priorda(vitb+vitb)   | **9.11** | **5.31** | **3.40** | **2.28** |
>
> **Performance Comparison on ScanNet (Abs Rel, ↓)**
> |        Setting        |   5   |  10   |  20  |  50  |
> |:--------------------:|:-----:|:-----:|:----:|:----:|
> | OMNI-DC              |15.57  |11.27  | 7.50 | 4.15 |
> | Marigold-DC          |11.81  | 6.68  | 4.24 | 2.78 |
> | PromptDA             |26.36  |19.99  |17.15 |15.40 |
> | DepthLab             |19.81  |14.21  | 9.78 | 6.49 |
> | priorda(vitb+vitb)   | **8.17**  | **5.02**  | **3.29** | **2.34** |
>
> These results strongly demonstrate our method's superior performance and robustness, even in the challenging extremely sparse settings.
>
> ---
> ## W2: Wrong Hyperlinks / Formatting of the paper
>
> We apologize for the oversight regarding the misaligned hyperlinks and text overlays. We address these issues in the updated manuscript.
>
> ---
> ## Q1: Robustness to extreme sparse or noisy data
>
> ### For extreme sparse data:
>
> As shown in our detailed response to Weakness 1 (W1), our method demonstrates impressive robustness and efficiency even under extremely sparse prior information, significantly outperforming competitive baselines. This performance is a core design feature of our two-stage alignment and refinement pipeline.
>
> ### For extreme noisy data:
>
> The "noise" in depth measurements primarily means blurry edges and missing details, but the overall metric scale is reliable. This numerical reliability of the metric scale is the fundamental motivation behind prior-based depth estimation. Therefore, the primary objective of our method is to fix the detail errors (noise) while keeping the overall scale from the prior.
>
> The robustness of our model against noise in depth measurements is evidenced by the superior performance on ARKitScenes and RGB-D-D (Table 4) and the visualiations (Figure 4-10). If the metric information in the prior is fundamentally flawed (e.g., random initialization), the prior-based depth estimation methods, including both ours and baselines, can not recover meaningful metric depth.
>
> ---
> ## Q2: Recursive Refinement via Conditional MDE
>
> We tested recursively feeding the conditional MDE's output back into itself. This idea does not bring further improvement and causes performance to degrade.
>
> | Iterations | abs_rel |
> |-------|---------|
> |   1   |  1.87   |
> |   2   |  2.01   |
> |   3   |  2.17   |
> |   4   |  2.31   |

---

### Author Response · Authors · 2025-12-04
**Summary of the Rebuttal Phase**

We sincerely thank the reviewers and the Area Chair for their valuable time and constructive feedback. This summary details the discussion process and our responses, aiming to provide a clear and transparent reference for the revision.

---
### **Reviewer Zgrm (Score: 8 $\to$ 8)**

* **Main Concerns:**
    1.  Latency and performance of our method in extreme conditions.
    2.  Feasibility of recursive refinement.
* **Our Response:**
    1.  We provided results demonstrating our method's significant advantage in extreme-case latency and performance compared to baselines.
    2.  We provided experimental results confirming that recursive refinement does not work.
* **Outcome Summary:** The reviewer highly praised the **novelty, extensive experiments, and strong performance, maintaining the score of 8.**

---
### **Reviewer thPV (Score: 2 $\to$ ?)**

* **Main Concerns:**
    1.  Low Preformance: The method underperforms in a significant portion of the evaluations.
    2.  Limited Novelty: The method primarily predicting per-pixel scale and shift to calibrate pixels.
* **Our Response:**
    1.  We provided average rank statistics demonstrating the overall comprehensive advantage of our method.
    2.  We clarified the misunderstanding: the first stage **computes, it does not "predict"**, the scale/shift, and its motivation is to **bridge the domain gap, not "calibrate pixels."** We emphasized that the core novelty is the **unified approach to handling any prior**, which the reviewer seemed to overlook.

* **Outcome Summary:** The reviewer acknowledged our explanation regarding performance, **indicating that the major concern about low performance may have been resolved.** We believe our clarification addresses the novelty misunderstanding, although they did not have an opportunity to reply further and modify the score.

---
### **Reviewer uZ3K (Score: 6 $\to$ 6)**

* **Main Concerns:**
    1.  Detailed inference efficiency.
    2.  Need for more ablation studies.
* **Our Response:** We provided more detailed experimental analysis and/or pointed to the corresponding locations in the manuscript.
* **Outcome Summary:** The reviewer confirmed that their concerns were addressed and **maintained the initial positive rating**.

---
### **Reviewer pbPR (Score: 6 $\to$ 6)**

* **Main Concerns:**
    1.  More experimental details.
    2.  Clarity of figures and presentation.
    3.  Results on additional benchmarks.
* **Our Response:**
    1.  We provided more detailed explanations and clarifications.
    2.  We updated figure captions for better clarity.
    3.  We included results on additional benchmarks, further illustrating our method's advantages.
* **Outcome Summary:** The reviewer stated that **many of their concerns were addressed** by the clarifications.

---

### Meta-Review · Area_Chair_GXkD · 2026-01-05

**Summary:**

In the initial review process the paper received primarily positive reviews (8 / 6 / 6 / 2). For the positive reviews, the concerns primarily centered on iterative improvements to the paper such as having explicit inference / latency measures as well as some additional ablations or benchmarks. For the negative review, the reviewer concern centered on poor performance and limited novelty.

**Reviewer Concerns:**

For all the positive reviews, the authors added content and addressed their concerns in the rebuttal. For the negative review, the authors provide additional proof that their method performs well (computing the average rank) and explains that the novelty of the paper having a comprehensive / unified approach to including the prior in a system used for depth computation. I believe that all of the concerns were addressed.

**Reviewer Scores:**

Likely, all the positive reviews would have maintained their score and the negative review may have increased from 2 to 6 since their concerns, I believe, were convincingly addressed by the authors.

---

### Decision · Program_Chairs · 2026-01-26

Accept (Poster)